# Polycage membranes for precise molecular separation and catalysis

Xiang Li [1,2], Weibin Lin [2,3], Vivekanand Sharma[2,3], Radoslaw Gorecki[1,2], Munmun Ghosh[2,3], Basem A. Moosa [2,3], Sandra Aristizabal [1,2], Shanshan Hong[1,2], Niveen M. Khashab [2,3] ✉ & Suzana P. Nunes [1,2,3,4] ✉

The evolution of the chemical and pharmaceutical industry requires effective and less energy-intensive separation technologies. Engineering smart materials at a large scale with tunable properties for molecular separation is a challenging step to materialize this goal. Herein, we report thin film composite membranes prepared by the interfacial polymerization of porous organic cages (POCs) (RCC3 and tren cages). Ultrathin crosslinked polycage selective layers (thickness as low as 9.5 nm) are obtained with high permeance and strict molecular sieving for nanofiltration. A dual function is achieved by combining molecular separation and catalysis. This is demonstrated by impregnating the cages with highly catalytically active Pd nanoclusters (~ 0.7 nm). While the membrane promotes a precise molecular separation, its catalytic activity enables surface self-cleaning, by reacting with any potentially adsorbed dye and recovering the original performance. This strategy opens opportunities for the development of other smart membranes combining different functions and well-tailored abilities.

Separation processes in the chemical and pharmaceutical industry, such as evaporation and distillation, are energy-intensive[1–3]. Membrane technology does not necessarily involve thermal transitions and can be an effective alternative for molecular fractionation and purification, particularly of systems sensitive to temperature. The selectivity of the membrane will dictate the number of steps required for an effective process. While commercial membranes have excellent performance in the separation of salts from water in large-scale desalination plants, they lack selectivity for molecules of similar size and charge. Interfacial polymerization (IP) has been one of the most successful strategies for the fabrication of nanofiltration and reverse osmosis membranes[4–9]. By this method, an ultrathin selective layer is formed on a porous support, by reacting classical monomers such as m-phenylene diamine (MPD) dissolved in water in contact with solutions of acid chlorides (e.g., trimesoyl chloride, TMC) in a nonpolar-organic solvent. Large areas of membranes can be in this way fabricated with a relatively small quantity of monomers. Although densely cross-linked, there is a random distribution of paths for transport with different sizes in the sub-nanometer range. We believe that strict selectivity can only be achieved by using building blocks of preformed structure with precise free volume for selective permeation. With that in mind, our group previously explored the fabrication of membranes by IP using macrocycles[10] such as amine-functionalized cyclodextrin[11] and trianglamine[12] as monomeric units, recently further explored by ref. 13. These highly cross-linked systems have much higher uniformity of "pores" for transport. In this work, we identify porous organic cages (POCs)[14–16] synthesized by means of dynamic imine chemistry as a promising molecular platform for membrane design, with the perspective of providing even more selective and preformed tuned paths for permeation. The motivation is to profit from the highly defined 3D structures of the cages as building blocks, as potentially advantageous in comparison to the rather 2D units of macrocycles that we previously investigated.

[1]Environmental Science and Engineering Program, Biological and Environmental Science and Engineering Division (BESE), Thuwal, Saudi Arabia. [2]Advanced Membranes and Porous Materials (AMPM) Center, Thuwal, Saudi Arabia. [3]Chemistry Program, Chemical Engineering, Physical Science and Engineering Division (PSE), Thuwal, Saudi Arabia. [4]King Abdullah University of Science and Technology (KAUST), 23955-6900 Thuwal, Saudi Arabia. ✉ e-mail: niveen.khashab@kaust.edu.sa; suzana.nunes@kaust.edu.sa

Besides the tuned permeation paths, cages can offer the possibility of adding multifunctionality to the membranes. Multifunctional membranes could have photoresponsivity[3,17,18], pressure[19], and thermal responsivity[20], and catalytic activity. Other monomers are under investigation[11,12,21–23] for multifunctionality purposes. We also reported smart covalent organic networks (CONs) with light-switchable pores for molecular separation prepared by IP[18]. A key factor for the practical production of thin-film composite (TFC) membranes is the reaction time. For production in continuous machines, a fast reaction is essential. COF approaches are frequently based on slow dynamic Schiff base chemistry[24–29], which might require more than 24 h to complete the film formation.

Combining POC structures and interfacial polymerization fulfills the requirement of a fast reaction with intrinsic functionalities. POC properties have been fine-tuned for target-specific applications, such as bioimaging[30], rare gas recovery[31], isotope hydrogen[32,33], and xylenes[34] and alkane/alkene separations[35] as adsorbents. Preliminary strategies for using POCs in the membrane field consisted of their incorporation in the form of mixed matrix membranes[36,37] and the deposition or crystallization of a pure POC layer[38,39]. Recently, a smart and responsive crystalline POC membrane was prepared by interfacial crystallization[40], showing a graded molecule sieving due to its switchable pore apertures. By a similar strategy, cages were formed and crystallized by counter-diffusion on a porous anodic aluminum substrate to prepare membranes tested for ionic sieving with a superior selectivity for mono/divalent ions[41]. These results confirmed the potential of POCs for function-customization membranes. However, POC formation and crystallization require a long time. Furthermore, the scale-up of defect-free POCs membranes prepared by crystallization would be highly challenging. Interfacial polymerization has been recently performed with a mixture of cages and piperazine as a monomer for membrane fabrication tested for salts and dyes filtration[42]. However, as in the case of mixed matrix membranes, approaches using mixtures of monomers dilute the advantages that POC would bring to a membrane.

In this work, we propose the interfacial polymerization of functionalized POCs as an effective method of membrane preparation, leading to a fully cross-linked and highly selective polycage layer. We then demonstrated that highly permeable polycage TFC membranes can be successfully prepared for organic solvent nanofiltration (OSN). Two POCs structures were selected: tren-cage and reduced imine cage (RCC3). They belong to two major types of organic cages. The cage's imine bonds could be reduced to obtain amino-functionalized cages, which could then be used as IP monomers in the aqueous phase (Fig. 1).

By choosing two different structures, we first confirmed the transferability of the approach. Second, we expected advantages in different ways. Tren-cages have a trigonal-bipyramidal geometry and are expected to pack and polymerize more densely at the interface. On the other hand, RCC3-like cages have larger cavities, which could be used to encapsulate nanoclusters or other guests. In this case, through a procedure analogous to that applied for the tren-cage, a unique membrane with high catalytic activity was built by encapsulating highly catalytically active Pd nanoclusters (~0.7 nm)[43], a concept that could be extended to a wide range of applications.

An inevitable issue in applying this concept was the solubility of the cages, since most of them have a large molecular weight and phobic moieties, such as phenyl. POCs that can dissolve in high concentrations in water are not reported yet, to the best of our knowledge. We propose an effective method of solubilization of POCs in an aqueous medium and use them to prepare TFC membranes by IP. Additives such as inorganic acids or trifluoroethanol (TFE) were key to solubilizing the cages in water.

## Results

### Preparation of polycage nanofilms (PCN) and TFC membranes therefrom

RCC3 and tren-cages with amino groups were synthesized as previously reported[44,45]. Their structure was confirmed by proton and carbon Nuclear Magnetic Resonance ($^1$H-NMR and $^{13}$C-NMR), as shown in Supplementary Figs. 1–6). The cage solubility in an aqueous phase is essential for the success of the IP approach. After a series of preliminary trials, two strategies to extend and secure the solubilization were proposed and implemented: (i) dissolving the amino-functionalized cage into an aqueous solution after adjusting the pH to 6.6 and (ii) dissolving it into a 1,2,3-trifluoroethanol (TFE)-H$_2$O mixture solution without pH adjustment (Supplementary Fig. 7a). A highly cross-linked ultrathin polycage film is formed (Supplementary Fig. 7b), once the clear solution contacts the organic phase (e. g. hexane), containing TMC (Fig. 1a). The polycage TFC membranes for performance evaluation were directly prepared on a porous polyacrylonitrile (PAN) support (Supplementary Fig. 8). While industrial TFC membranes for desalination are frequently prepared by IP on polysulfone supports, PAN is a better choice for OSN, due to its superior stability in solvents. The preparation of PAN support secures the mechanical stability of the membrane and facilitates the future scale-up in continuous machines. RCC3 or tren-cages were used as monomers, as specified in plots and tables, followed by an indication of the aqueous phase (acid or TFE added) and the reaction time in

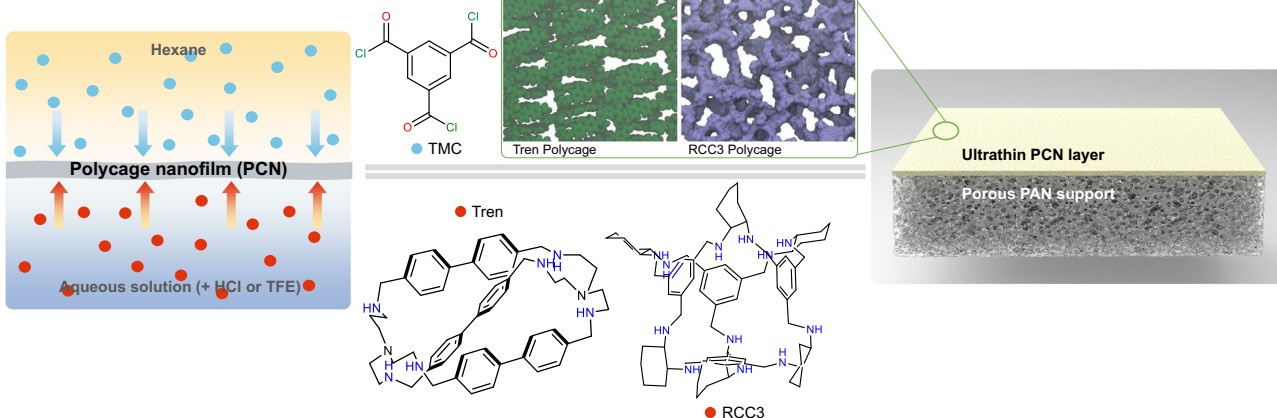

**Fig. 1 | Synthesis of polycage thin-film composite membrane by interfacial polymerization (IP).** TMC (blue dots) dissolved in an organic phase (e.g., hexane), amino-functionalized cages (tren-cage or RCC3, red dots) dissolved in an acidic aqueous solution or TFE-H₂O solution, and schematic cross-linked network of tren-cage and RCC3-based layers prepared by IP on asymmetric porous support (in a green rectangle).

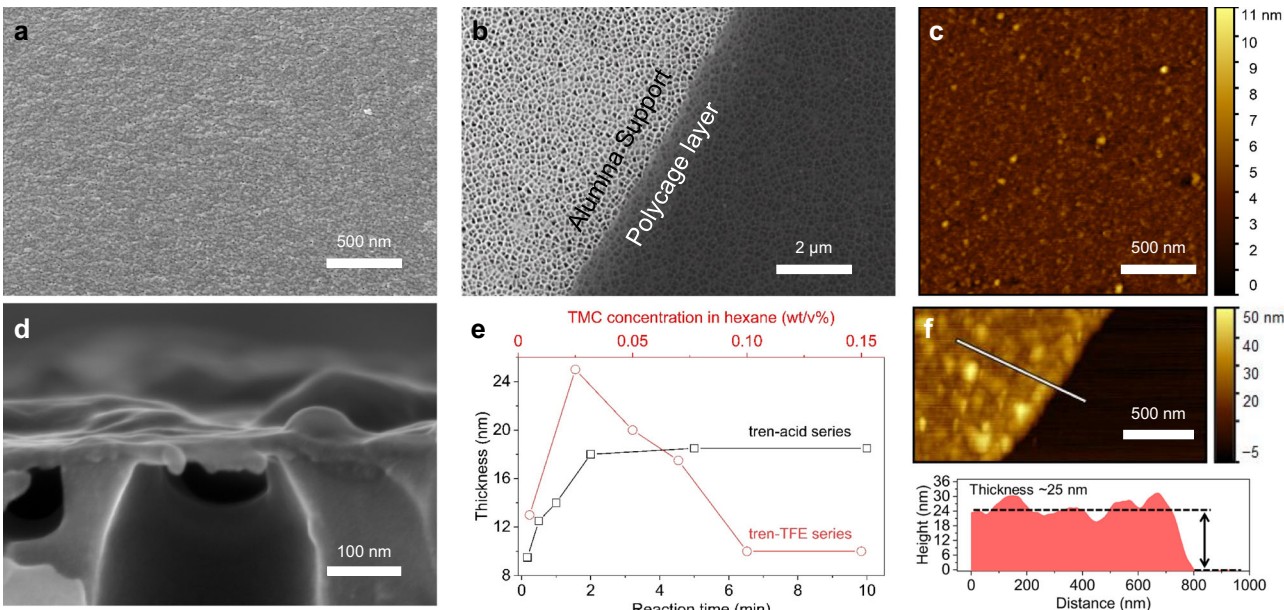

**Fig. 2 | Morphology of interfacial polymerized polycage thin films. a** SEM image of the surface of a nanofilm prepared by IP on a PAN membrane support. **b** SEM image of a free-standing nanofilm collected on an alumina porous support; **c** AFM image of the nanofilm (**b**). **d** Cross-sectional SEM image of the nanofilm (**b**) collected on an alumina support. The nanofilm was prepared by IP of tren-cage (0.05 wt%) in an aqueous solution (pH = ~6.6) with TMC (0.1 wt/v%) in hexane and a reaction time of 10 min. **e,** Thickness of tren-acid series and tren-TFE series films (measured by AFM) as a function of the reaction time and TMC concentration in hexane, respectively. **f** AFM image and corresponding height profile of a section of a tren-TFE-0.025 nanofilm on top of a silicon wafer. A scratch was made to expose the wafer surface and allow the measurement of the height from the silicon wafer surface to the upper nanofilm surface.

parenthesis. The TMC concentration in hexane was specified, if different than 0.1 wt/v%. The pH adjustment is an effective approach to facilitate the reaction and has been applied in previous works of our group[12]. The solubility of amino-functionalized cages in aqueous solution varies with their size and hydrophobicity. The maximum concentration of tren-cages in water is about 0.05 wt% after optimization at room temperature, while RCC3 could be dissolved in concentrations higher than 1 wt%. The scanning electron microscopy (SEM) image of the tren-acid (reaction time 10 min) membrane shows a continuous and smooth surface without visible defects (Fig. 2a and Supplementary Figs. 8–10). A very thin layer is formed, as observed by SEM (Supplementary Figs. 11–12).

For selected chemical and morphological characterizations, free-standing layers were prepared and collected on substrates, as described in Supplementary Fig. 13a. Figure 2b shows the tren-acid (reaction time 10 min) layer collected on a porous alumina support. Its transparency is high, indicating a very low thickness. The AFM image in Fig. 2c confirms a thickness of 18 nm and indicates a low roughness of 4 nm. The AFM topological profile is shown in Supplementary Fig. 14. In the case of membranes prepared with TFE/H$_2$O as polar phase (strategy ii), a free-standing tren-TFE film is visibly formed at the hexane/TFE-H$_2$O interface, using 0.3 wt/v% in TFE-H$_2$O mixture with a volume ratio of 1.5 (Supplementary Fig. 7b). The observations on free-standing systems guided the optimization of conditions needed for a translation into the preparation on PAN supports (Supplementary Fig. 13b). The SEM images of tren-TFE or RCC3-TFE membranes (Supplementary Figs. 15, 16a, b) directly prepared on a PAN support confirmed a full coverage, which sufficiently proves the feasibility of the strategy ii also in processes that could be scalable to continuous machines (Supplementary Fig. 13c).

A series of relevant conclusions were obtained based on experiments with free-standing films. We found that the thickness of the polycage layer could be finely tailored by two optional strategies. In Fig. 2e, the thickness of free-standing tren-acid and tren-TFE films in response to reaction time and TMC concentration, respectively, have a

different trend. In the former case, by increasing the reaction time from 10 s to 5 min, the thickness of the tren-acid film increases from 9.5 to 18 nm (Supplementary Fig. 14). But a longer reaction time did not lead to a thicker film. Similarly, the thicknesses of RCC3-acid films increase from 6.5 nm and 12 nm by increasing the reaction time from 10 s to 10 min (Supplementary Fig. 18). In the case of tren-TFE, the reaction finishes within 10 s at TFE-H$_2$O/hexane interface, without further changes. Then, the effect of the TMC concentration in hexane was primarily investigated. By increasing the TMC concentration from 0.005 wt/v% to 0.025 wt/v%, the thickness of tren-TFE films increases from 13 up to 25 nm (Fig. 2f and Supplementary Fig. 19). The thickness of tren-TFE films goes down to 15 nm again when the TMC concentration increases up to 0.1 wt/v%. This behavior is confirmed by the SEM observations in Supplementary Fig. 20. The use of a higher TMC concentration doesn't result in the formation of a thinner film (Supplementary Fig. 19e, f). Ultrathin RCC3-TFE films with a thickness of around 23 nm were also synthesized using 0.5 wt/v% in a TFE-H$_2$O mixture with a volume ratio of 1 (Supplementary Fig. 16c).

An important consideration to understand the thickness variation is the following. The solubility of n-hexane in TFE is 2.8 v/v%, and the solubility of TFE in hexane is around 1.3 v/v%[46] Compared with the H$_2$O/hexane interface, the initial reaction zone is wider. The surface tension of TFE is 21.1 mN m$^{-1}$, much lower than that of water (72.75 mN m$^{-1}$). The interface is much more diffuse and does not stop the monomer transport as the polymer layer is being formed. In the beginning, an increase of TMC concentration from 0.005 to 0.025 wt/v% promotes the diffusion of TMC from the hexane phase to the H$_2$O-TFE phase and leads to thicker films. When the TMC concentration continues to increase up to 0.1 wt/v%, the ratio of TMC to an amine-functionalized cage close to the interface increases and a thinner film is formed. The film thickness is then limited by the incipient film formation itself. An even higher concentration (0.15 wt/v%) does not further lead to a thinner film.

The high-resolution transmission electron microscopy (TEM) images reveal the amorphous structure of the polycage films, as no

ordered crystalline areas are present (Supplementary Fig. 21). The lack of crystallinity is confirmed by wide-angle X-ray diffraction (XRD) (Supplementary Fig. 22).

Solvent-resistant, defect-free cross-linked cage layers thinner than 25 nm were formed by dissolving the cages in (i) $H_2O$ at pH 6.6 and (ii) TFE-$H_2O$. Comparing these two strategies, a clear difference in diffusion rates between monomers from the aqueous and organic phases should be expected. As shown in Supplementary Fig. 23, the diffusivity of the amino-functionalized cage into the organic phase is extremely slow under strategy i, due to the ionic feature of the amine-functionalized cage and its large size compared to classical amines

(e.g., MPD). But under strategy ii, the cage diffuses faster into the hexane phase from the TFE-$H_2O$ mixture side.

## Membrane chemical characterization

The chemical structure of polyamide in polycage nanofilms was analyzed by ATR-FTIR spectra without the interference of the polymer substrate (Fig. 3a and Supplementary Fig. 24). The peak of –NH- stretch at 3290 cm$^{-1}$ in the spectra of tren-cage and RCC3 is greatly attenuated in their nanofilms. Furthermore, the skeletal –C=O bonds form the secondary amide group whose peak appears at 1631 cm$^{-1}$ [47]. These results clearly verify the formation of the polyamide nanofilms via

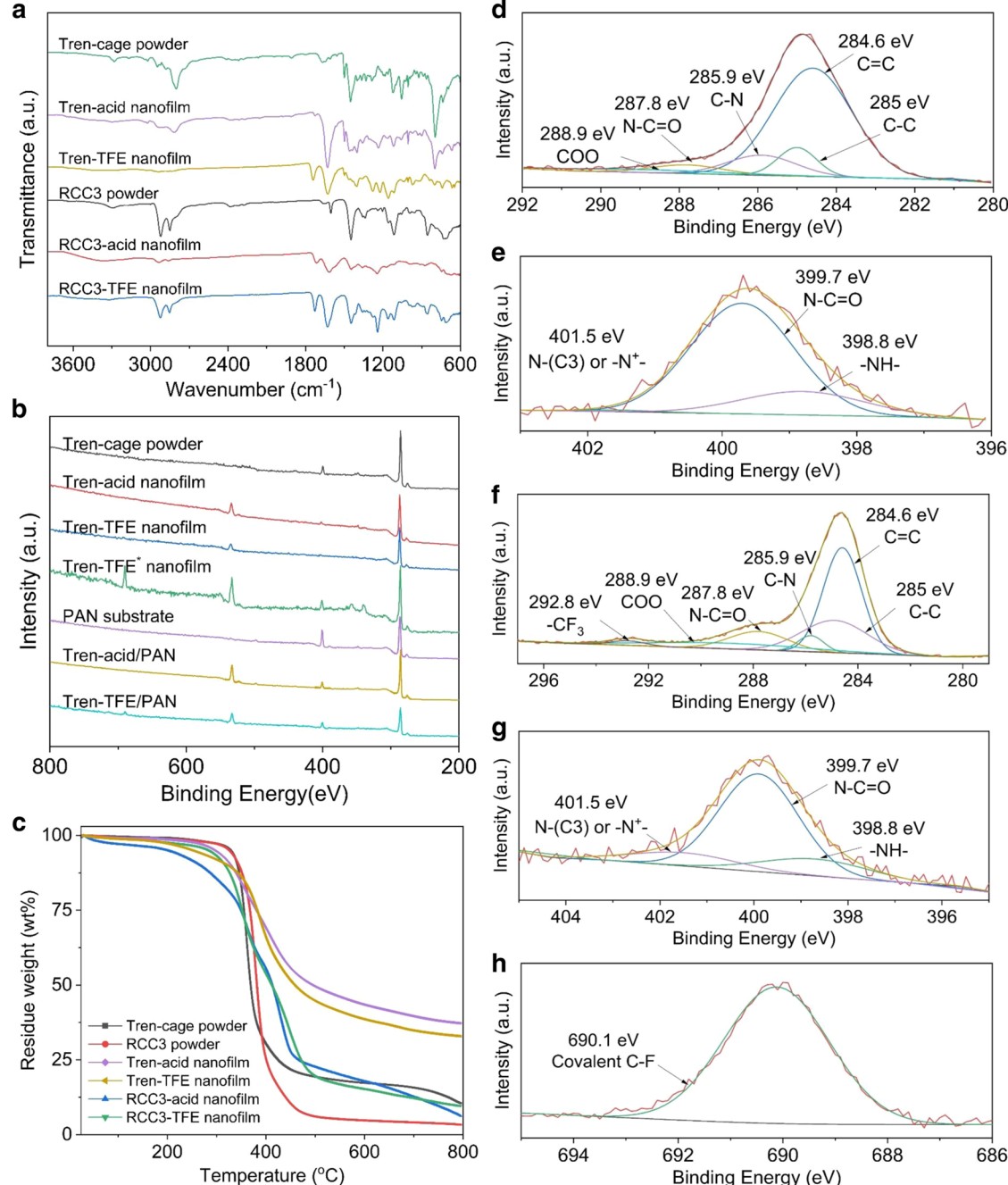

**Fig. 3 | Chemical characterization of monomers and nanofilms. a** ATR-FTIR spectra of tren-cage powder (in green), RCC3 powder (in black), tren-acid nanofilm (in pink), tren-TFE nanofilm (in yellow), RCC3-acid nanofilm (in red), and RCC3-TFE nanofilm (in blue). **b** XPS survey spectra of PAN substrate (in pink), tren-cage powder (in black), tren-acid nanofilm (in red), tren-TFE nanofilm (in blue), tren-TFE nanofilm* (in green), tren-acid/PAN (in yellow), and tren-TFE/PAN (in cyan) composite membranes. **c** TGA curves of pristine cages and their PCN films. **d** XPS C1s spectra and **e** N1s spectra of the tren-acid nanofilm. **f** XPS C1s spectra, **g** N1s spectra, and **h** F1s spectra of tren-TFE nanofilm.

polycondensation between TMC and amino-functionalized cages. Additionally, there are minor changes of O=C–N bending in-plane and C–N stretching at 750, 1260, and 1400 cm$^{-1}$ in the spectrum, respectively, after the cross-linking occurs. Also, the C=O stretching of carboxylic acids in nanofilms clearly appears at 1720 cm$^{-1}$, revealing a pendant carboxylic acid groups after interfacial polymerization. There is a relatively strong adsorption peak of the CF$_3$ groups at 837 cm$^{-1}$ in a tren-TFE nanofilm rather than a tren-acid nanofilm, demonstrating the existence of a side reaction between TMC and TFE at hexane/TFE-H$_2$O interface. Furthermore, there is an overall similarity between the spectra of both nanofilms and amino-functionalized cages, indicating that the cage skeletons remain intact. To further determine the cross-linking degree of the nanofilms, X-ray photoelectron spectroscopy (XPS) was performed (Fig. 3b, d–h and Supplementary Figs. 25, 26, and Supplementary Table 1). When preparing the polycage layers on PAN support, considering the case of tren-cage, the O/N ratio of a tren-acid membrane increases from 0.05 to 2.1 compared to the PAN support. The existence of the F element in tren-TFE is also evident proof of the side reaction of TFE. The higher O/N ratio and F content suggest that the surface of the PAN support is covered by a PCN layer. Also, there is a similar tendency when RCC3 was utilized. However, to avoid the interference of the PAN substrate (Supplementary Figs. 26a, b), free-standing polycage nanofilms were analyzed to precisely clarify their chemical structure. As a comparison, functionalized cage powders were also analyzed. As for both functionalized cages, there exists a main peak at 398.8 eV in the N1s spectrum, assigned to the –NH– (Supplementary Fig. 26d, f). The main peak shifts to 399.7 eV in polyamide nanofilms (Fig. 3e and Supplementary Fig. 26h), which corresponds to the N–C=O, and the peak of –NH– greatly decreased[48,49]. Therefore, the percentage of the reacted –NH– group per tren-cage molecule could be calculated according to the ratio of the peak of N–C=O (399.7 eV) to the whole signal. Consequently, 40 and 61% of the –NH– groups were consumed within the formation of the tren-acid and RCC3-acid nanofilm by IP, respectively. The incomplete reaction of –NH– groups in cages is limited by their large size and high steric hindrance. On the other hand, the much lower concentration of tren-cage in an aqueous solution than RCC3 amino-functionalized cages greatly reduces the diffusion rate. Furthermore, in the case of PCN-acid nanofilms, the deconvolution of the C 1 s narrow scan spectrum identifies five-component peaks: C=C (284.6 eV), C–C (285 eV), C–N (285.9 eV), N–C=O (287.8 eV), and COO (288.9 eV) (Fig. 3d and Supplementary Fig. 26g)[6,50]. Among them, the N–C=O and COO are only related to the TMC segments. But in the case of PCN-TFE nanofilms, characteristic peaks of covalent C–F (690.1 eV) and –CF$_3$ (292.8 eV) are also identified in the F1s[51] (Fig. 3h and Supplementary Fig. 26k) and C1s (Fig. 3f and Supplementary Fig. 26i) spectrum of tren-TFE and RCC3-TFE nanofilm, respectively. It's also calculated from Fig. 3g and Supplementary Fig. 26j that 64 and 58% of the –NH– groups reacted with TMC for tren-TFE and RCC3-TFE nanofilms, respectively. We found that compared with common amines, such as MPD, their cross-linking densities are lower, due to their unique steric structures. Nevertheless, the full crosslink of cages with 6 and 12 reactive groups is not necessary to provide nanofilms with excellent stability in a wide range of solvents.

As for the stability, TGA measurements of free-standing films indicated that they are thermally stable up to 300 °C (Fig. 3c). There is a sharp weight decrease ranging from 300 to 460 °C in the case of tren nanofilms, while from 300 to 470 °C in the case of RCC3 nanofilms, which correspond to amino-functionalized cages moiety. According to the curves, we can estimate amino-functionalized cage contents are around 70 and 80 wt% in tren and RCC3 nanofilms, respectively. It indicates that a lower amount of TMC as a cross-linker is required to fuse large cages, which is consistent with the cross-linking density analysis. The surface hydrophobicity of the membranes was verified, as the contact angle was close to 90° (Supplementary Fig. 27). The PAN substrate has a low water contact angle of 43°. Based on the analysis of their chemical composition, a high proportion of nonpolar groups, including cyclohexyl and phenyl of cages skeletons, contributes. The introduction of fluorine to the polycage network when the IP reaction is conducted using TFE-H$_2$O must be considered and this could be beneficial for the transport of organic solvents through membranes.

## Performance of PCNs/PAN TFC membranes

The performance of four series of polycage TFC membranes directly prepared on PAN supports was evaluated at 20 °C. The permeance of methanol and water and the rejection curves of polycage membranes are affected by the synthesis conditions, such as reaction time and TMC concentration. For example, the methanol permeances of tren-acid membranes in Fig. 4a sharply decrease from 9.5 to 3.8 L m$^{-2}$ h$^{-1}$ bar$^{-1}$ when the reaction time increases from 10 s to 5 min, respectively. The further increase in reaction time does not lead to an additional change in the methanol permeance. The water permeance has the same trend. The predominant factor is the thickness change with reaction time. This could be clearly confirmed in experiments with free-standing films by AFM. An additional factor could be a change in the cross-linking degree happening at the same time. To evaluate a potential alteration of the cross-linking degree of tren-acid membranes, we measured by XPS the content and kind of bond of N in different membranes, comparing tren-acid (reaction time 30 min) and tren-acid (reaction time 10 s), as shown in Supplementary Fig. 26l, m. Based on the N1s spectra in both samples, the percentage of the reacted –NH– group per tren-cage molecule could be calculated according to the ratio of the peak of N–C=O (399.7 eV) to the whole signal. 40 and 34% of the –NH– groups were consumed during the interfacial polymerization and formation of the tren-acid (reaction time 30 min) and tren-acid (reaction time 10 s) layers, respectively. Therefore, we conclude that a shorter reaction time leads to a membrane with a lower cross-linking degree, and this affects the permeance as well. The fact that the methanol permeance is higher than that of water can be explained by the lower viscosity and by a better interaction between methanol and the hydrophobic phenyl moiety of the tren-cage in the membrane. RCC3-acid membranes have much higher liquid permeances. Specifically, the RCC3-acid (reaction time 10 s) membrane has high methanol and water permeances of around 53.0 and 37.5 L m$^{-2}$ h$^{-1}$ bar$^{-1}$, respectively. The possible reason for the high permeance is first the low thickness of the polycage layer, which is equivalent to packing only about seven layers of RCC3 cages. Both methanol and water permeances decrease as the reaction time increases, similar to what was observed for tren-cage membranes.

Figure 4b, f show the polycage-acid membrane selectivity for a range of dyes. All tren-acid membranes exhibit almost complete rejection (>99%) of dyes with a molecular weight larger than 580 g mol$^{-1}$. For dyes of lower molecular weight, such as methyl orange, the rejection was not higher than 65%. For the RCC3-acid membranes, prepared with a reaction time longer than 30 s, the rejection curves were much less sharp. The range of minimum molecular weight with 90% rejection is broader and more dependent on the reaction time. The rejection becomes constant at a value >97 % above a molecular weight of 700 g mol$^{-1}$.

The differences between tren- and RCC3-based membranes are evident. There are three potential reasons for that: (i) the transport channels of the RCC3 cage (diameter >0.7 nm) are larger than those of the tren-cages (<0.5 nm) (Supplementary Fig. 28); (ii) compared to the larger [4 + 3] tetrahedral-like RCC3 cage, the ellipsoidal tren-cage can more efficiently pack and reduce non-selective voids (Supplementary Movies 1, 2); (iii) a more effective cross-linking might occur for the tren-cage than for RCC3 in the narrow and confined interface. Besides the cage structure, the membrane preparation conditions affect the cross-linking and, therefore, the performance. Specifically, it is necessary to improve the rejection ability of RCC3-acid membranes through post-heat treatment (Supplementary Fig. 29).

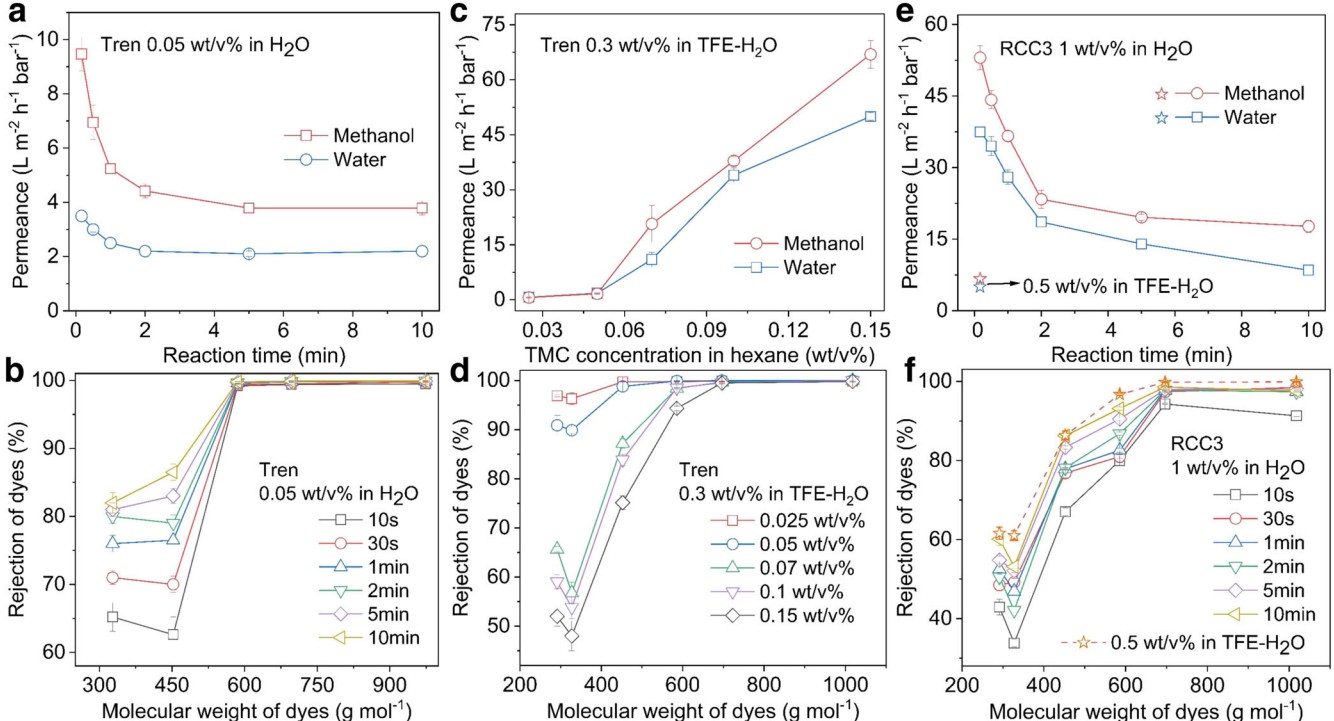

**Fig. 4 | Thin-film composite membranes performance. a**, **c**, **e** Pure solvent permeances for methanol and water through PCNs nanofilm composite membranes on PAN supports. Nanofiltration was conducted in a dead-end filtration system at 20 °C under 1 bar. **b**, **d**, **f** Rejection versus molecular weight of dyes: Rose Bengal (RB, 974 g mol⁻¹); Congo Red (CR, 697 g mol⁻¹); Acid Fuchsin (AF, 586 g mol⁻¹); Orange G Sodium (OGS, 452 g mol⁻¹); Methyl Orange (MO, 327 g mol⁻¹); and Methyl Red (MR, 269 g mol⁻¹) in methanol. Nanofiltration tests were separately conducted for each dye in a dead-end filtration system at 20 °C under 1 bar. Four kinds of PCNs/PAN membranes were prepared: in **a** and **b** 0.05 wt% tren-cage solution in H₂O and 0.1 wt/v% TMC solution in hexane; in **c** and **d** 0.3 wt/v% tren-cage in TFE-H₂O (volume ratio, 1.5) solution, 10 s of reaction time and different TMC concentrations; in **e** and **f** 1 wt% RCC3 solution in H₂O and 0.1 wt/v% TMC solution in hexane, and RCC3-TFE-0.005/PAN membrane was prepared with 0.5 wt/v% RCC3 in TFE-H₂O (volume ratio, 1) solution, 0.005 wt/v% TMC in hexane and 10 s of reaction time.

When keeping the amino-functionalized monomer structure and concentration constant in the aqueous phase, the TMC concentration in the organic phase affects the membrane characteristics, as in the case of classical membranes prepared with MPD as an amine monomer. As in the case of MPD, higher TMC concentration could lead to the formation of a thicker film with a higher cross-linking density. A higher TMC concentration leads to lower liquid permeances and better dye rejections in the case of MPD and the PCN-acid membranes (Supplementary Figs. 30a–d). The permeances and dye rejection curves of PCN-TFE membranes behave differently in response to the change in TMC concentration, as shown in Fig. 4c–f and Supplementary Fig. 30e–h. Specifically, during tren-TFE membranes preparation, the optimized range of TMC concentration lies between 0.025 and 0.15 wt/v%. Methanol permeances rapidly increase from 0.6 to 67 L m⁻² h⁻¹ bar⁻¹ when the TMC concentration increases from 0.025 to 0.15 wt/v%. On the other hand, both tren-TFE-0.025 and tren-TFE-0.05 membranes exhibit a high rejection (>90%) of dyes with a molecular weight larger than 290 g mol⁻¹. Furthermore, tren-TFE membranes prepared with a higher TMC concentration have very similar and sharp rejection curves, revealing the uniform cross-linked structure formed within this range of TMC concentration. But methanol permeances also increase to 29.7 L m⁻² h⁻¹ bar⁻¹, once the TMC concentration reduces to 0.001 wt/v% (Supplementary Fig. 30e). But as for tren-TFE membranes with TMC concentration between 0.2 and 0.5 wt/v%, the methanol permeances remain 95 ± 5 L m⁻² h⁻¹ bar⁻¹ and the rejection of AF dye remains about 65%. It reveals similar organic cross-linked structures generated within a 10 s reaction. This behavior of tren-TFE membranes is consistent with the thickness change of their corresponding free-standing tren-TFE films in Fig. 2e, which is derived from the difference in monomer diffusion rate and reaction site, as explained before. In the case of RCC3-TFE membranes, there is a

similar behavior (Supplementary Figs. 30g, h). We find that the optimum tren-TFE-0.005 membrane shows a similar rejection curve for dyes to the tren-acid-10 membrane, even though tren-TFE-0.005 is thicker with a similar cross-linking density. This might be attributed to the presence of fluorine. These measurements demonstrate the ability of tailoring PCN membranes, which outperform most commercial membranes or exhibit higher size-selectivity compared with other state-of-the-art membranes (Supplementary Table 2)[1,6,11,12,21,52–55]. For example, the tren-acid-10s membrane can precisely separate acid fuchsin (-100%) and orange G sodium (-60%).

Due to the well-defined transport paths provided by the cages, the membranes could be more effective in separations by size, shape, and charge than classical TFC membranes prepared with MPD. A minor effect that might be due to differences in molecular shape is repetitively seen in Fig. 4d, f: a minimum in rejection for MO (327 g mol⁻¹), while the rejection of MR is higher, even though the molecular weight is lower (269 g mol⁻¹). However, MR is slightly bulkier than MO. This effect was not observed in a control membrane prepared with MPD instead of cages (Supplementary Fig. 29b).

To further demonstrate the fine selectivity of the membrane based on molecular size and charge, a mixture of four neutral and four negatively charged dyes as markers was investigated with the tren-acid (reaction time 10 s) membrane (Supplementary Fig. 31a). Higher rejections of MO and OGS were detected, compared with analogous testing of single dye solutions in methanol. The rejection of DHBS dye with a molecular weight of 260 g mol⁻¹ was around 75%. It means that the interactions between solutes and the membrane should be considered when a real mixture for the separation purpose is investigated. Nevertheless, as for neutral markers, the membrane was highly selective, with almost full rejection (99.8%) of BrB (624 g mol⁻¹) and low rejection of C6 (350 g mol⁻¹). Besides that, MO and C6 with similar sizes

could be well separated based on their difference in charge. We also investigated a polyphenol mixture to mimic the purification and concentration of bioactive polyphenols abundant in plants. Supplementary Fig. 31b shows that the membrane exhibited low rejection (<50%) for polyphenols with less rigidity or non-planarity, and the rejection increased with the molecular weight. We believe a potential application could be the separation of chlorophyll (>653 g mol⁻¹) from bioactive polyphenols (flavonoids and ellagitannins, <350 g mol⁻¹) in plant extractants.

The permeations of various organic solvents through different polycage membranes are reported in Supplementary Fig. 32. It is evident that the best linear correlation is observed between permeation and the inverse viscosities in all cases. This indicates that viscous flow practically following the Hagen–Poiseuille equation is the dominant mechanism of transport with low permeance measured for viscous liquids like isopropanol and the highest permeances for hexane and acetone. The linearity is better followed for the tested RCC3-based membranes, which have the largest cavities for transport. The largest deviations from the pure viscous flow were observed for the tren-acid (reaction time 10 s) membranes, which also have the lowest permeance values, probably with the smallest pores. In this case, the thermodynamic interactions between solvents and the membrane material might play a more prominent role. However, a tentative plotting the permeance vs a combination of the inverse of viscosity and polar contribution to the Hansen solubility parameter, as previously adopted for other OSN systems, did not lead to a simple correlation. The solvent fluxes through the polycage membranes linearly increase with the applied pressure, reflecting a stable microstructure within the selective layer. The polycage membranes are also very stable after the immersion in various organic solvents (e.g., DMF) for 7 days (Supplementary Fig. 34a). The solvent permeance remains stable after long-term operation (Supplementary Fig. 34b), and the rejection of dyes

(acid fusion) was still the same with practically full rejection (Supplementary Fig. 34c).

## Molecular modeling of PCNs

Molecular simulations were performed to better understand and compare the different performances of the fabricated membranes with the microstructure network and free voids for permeation, and molecular simulations. Molecular modeling of tren and RCC3 cages was constructed considering the single crystal structures, optimized with ORCA package by density functional theory (DFT) calculations. The amorphous polymer models were generated using the Polymeric program, followed by a 21-step geometry relaxation procedure via the LAMMPS package. Simple expressions of tren and RCC3 were applied here for convenience to present the two different membranes prepared from tren and RCC3 cages, respectively.

As shown in Fig. 5a, b, the model representation for the membranes shows the structure of the cross-linked tren and RCC3 cages synthesized here as a distinct rigid 3D network with a density of 1.181 and 1.114 g cm⁻³. By inserting theoretical probes of 1 Å radius, the highlighted blue areas in Fig. 5a, b are presented, indicating a large fraction of interconnected free volume accessible to the probes in both tren and RCC3 membranes. The distribution of voids with different sizes and scales is shown in Fig. 5c, d. Each color corresponds to the largest probe radius that could be inserted. Compared with a scale from 1.40 to 2.40 Å of tren in Fig. 5c, RCC3 membranes have a larger void scale of up to 2.70 Å. Consequently, the relatively larger void sizes and scales render the RCC3 membrane higher liquid permeance, but poorer rejection ability. As shown in Fig. 5e, f, a simple plot of the voids size distribution can be easily derived from the simulation results, revealing that the most frequently present voids of tren have an average size of 3.0 Å, while RCC3 have a larger average size of 3.2 Å (Fig. 5c). The void size distribution of the RCC3 membrane is slightly

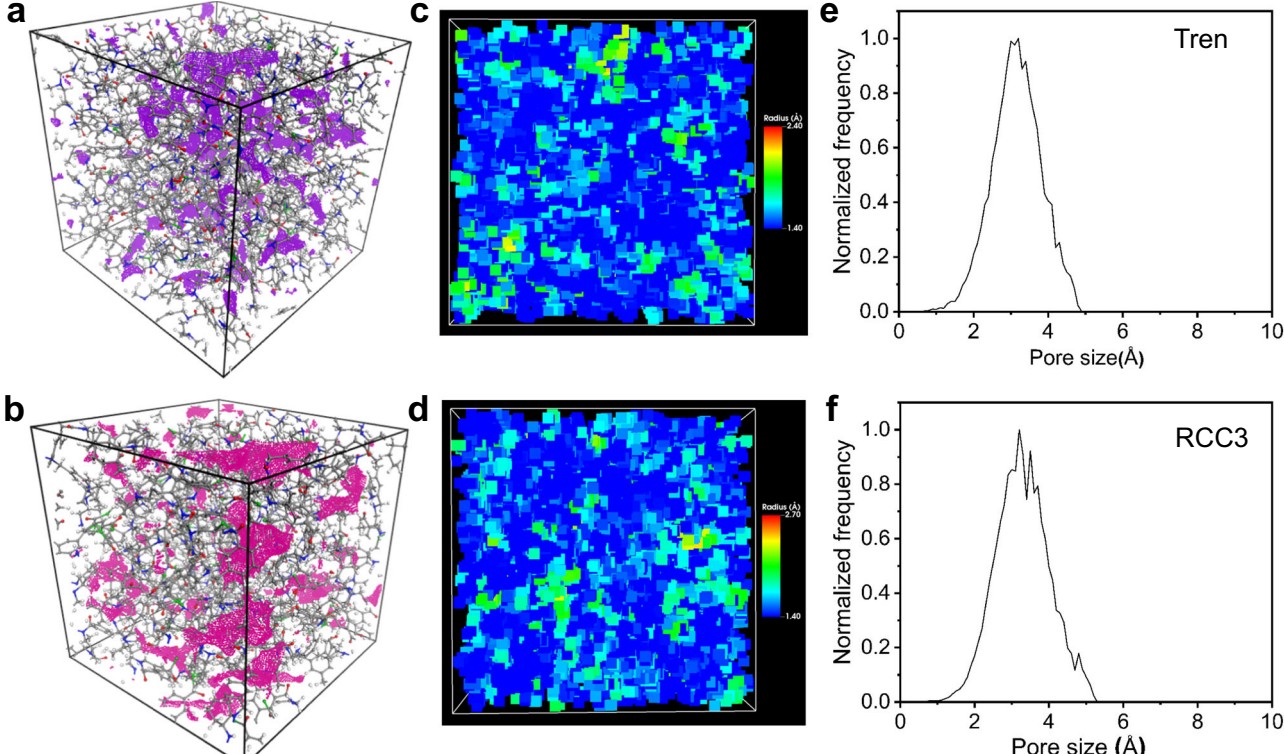

**Fig. 5 | Molecular modeling of the membranes.** Three-dimensional view of an amorphous cell of **a** tren membrane and **b** RCC3 membrane with a dimension of 35 Å × 35 Å × 35 Å, and the accessible surface at a probe radius of 1 Å marked violet

and magenta, respectively. Voids distribution of **c** tren membrane and **d** RCC3 membrane with size distinguished by color. Simulated pore size distributions of the fabricated **e** tren membrane and **f** RCC3 membrane.

broader than that of tren membranes, which results in a better rejection ability of small molecules and higher size-selectivity for dyes. Benefiting from the larger void size and interconnected microporosity, RCC3 membranes can have higher permeance for both polar and nonpolar solvents. Therefore, these simulated results about pore structures of membrane networks are consistent with their experimental permeance and size-selectivity performance.

The model simulations here could be interpreted as being exclusively related to the diffusivity of the cross-linked tren and RCC3 cage layers. The thermodynamic interactions between the permeant molecules and the membrane are not considered in these simulations. They could potentially influence the solubility and overall permeance, swelling the membrane, or altering the absolute sizes of cavities and pores. Nevertheless, the simulations reflect the moderate free volume and porosity of the membranes, proving the significance of the POC building blocks in creating permanently interconnected microporosity within the membranes for the diffusivity of permeants.

## Poly-Pd@RCC3 membrane with catalytic activity

The above results demonstrate the feasibility of the proposed strategies for the fabrication of polycage TFC membranes. The potential of POCs for membrane application can be brought to another level if additional functions could be introduced. A possibility we explored is the integration of catalytic clusters. In this way, membranes with double function, molecular separation, and catalytic activity, could be obtained. A recent review of approaches considering the encapsulation of functional guests inside POCs has been published by ref. 56. We chose to encapsulate highly catalytically active Pd nanoclusters inside RCC3 by a reverse double-solvents approach (RDSA)[43]. By encapsulation in cages, highly distributed catalyst nanoclusters are expected to be formed all over the selective layer. The confinement within cages should provide stability and the chosen system is claimed to be useful for various liquid-phase reactions. The HR-TEM analysis of the Pd@RCC3 revealed the uniform distribution of the Pd nanoclusters inside the cages. Only in part of the samples larger aggregates were observed (Supplementary Fig. 35a). All over the samples, well-distributed Pd nanoclusters were imaged (Supplementary Fig. 35b, c). By measuring the size of the smallest non-aggregated Pd nanoclusters (their crystalline structure makes them distinguishable from the amorphous carbon background), it was possible to estimate the size of the cages at ~0.7 nm (Supplementary Fig. 36).

The Pd@RCC3 cages were then used in IP reactions to produce thin catalytic membranes. A poly-Pd@RCC3 membrane was obtained using a 0.5 wt/v% Pd@RCC3 solution in TFE-$H_2O$ mixture, 0.01 wt/v% TMC solution in hexane, with a reaction time of 1 min, followed by drying for 12 h in the air. HR-TEM Micrographs of the PCN and poly-Pd@RCC3 films (and respective fast-Fourier transform (FFT) typical for crystalline structures) are presented in Fig. 6a–f. The structure of the poly-Pd@RCC3 film, in contrast to the completely amorphous structure of PCN film, had areas of distributed crystalline structures present, which is attributed to the Pd nanoclusters formed inside the

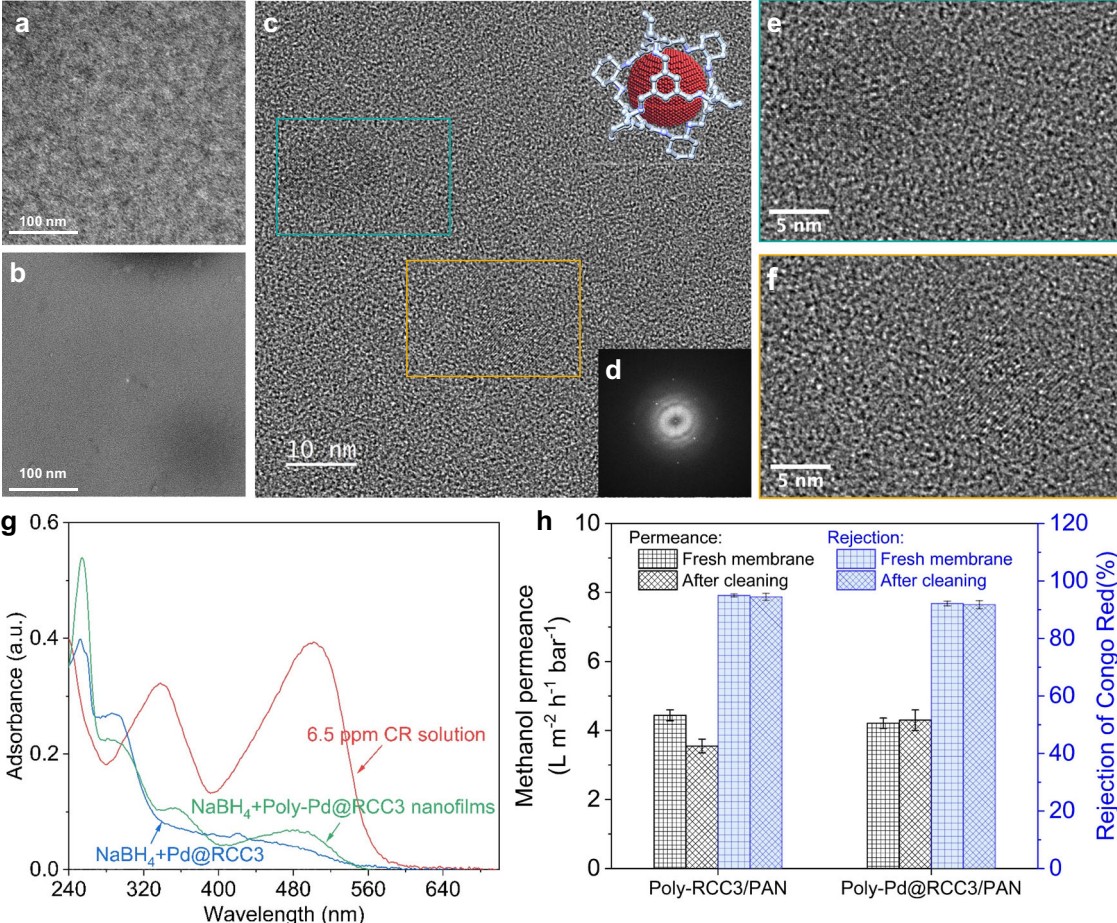

**Fig. 6 | Catalyst characterization and testing.** HR-TEM micrographs of PCN and poly-Pd@RCC3 films: **a** Amorphous PCN film, **b** Poly-Pd@RCC3 film having visible areas of higher (darker) and lower (lighter) concentration of Pd nanoclusters; **c, e, f** Lattice fringes of the Pd nanoclusters visible on the poly-Pd@RCC3 film, **d** FFT of the poly-Pd@RCC3 image which indicates the presence of crystalline structures within the film. **g** UV-Vis absorbance spectra for the reduction of Congo Red (CR) in the presence of the free-standing poly-Pd@RCC3 nanofilms or Pd@RCC3 catalyst at 293 K. **h** Self-catalytic cleaning property of poly-Pd@RCC3/PAN membrane.

cages and suggests how the cages are distributed within the active layer of the membrane.

The Pd clusters promote a fast catalytic reduction of organic dyes (MO and CR) within 60 s (Supplementary Fig. 37), following a catalytic process previously reported for encapsulated metals[43].

Supplementary Fig. 37 shows the OSN membrane performances. The poly-Pd@RCC3/PAN membrane has a lower methanol permeance and dyes rejections, compared with poly-RCC3/PAN membrane, possibly due to the occupation of the cavities inside RCC3 with Pd nanocluster, indirectly proving the important role of RCC3 cavity in molecular transport and sieving.

As expected, as-prepared poly-Pd@RCC3 membranes have catalytic activity. This is firstly demonstrated with free-standing poly-Pd@RCC3 nanofilms in Fig. 6g and Supplementary Movie 3. To avoid any contamination, the poly-Pd@RCC3 nanofilms were washed six times with methanol by sonication-centrifugation operation for use. In the Supplementary Movie, we see that the white nanofilms as catalyst became red in CR solution, and then turned back to white or gray with the addition of NaBH$_4$. Compared with the Pd@RCC3 catalyst, it took longer (about 2 min) to fully reduce CR because, in this case we have a heterogeneous solid-liquid reaction on the membrane surface, instead of the highly dispersed Pd@RCC3 in solution. After treatment with NaBH$_4$ in the presence of Pd@RCC3 or poly-Pd@RCC3 nanofilms, UV-Vis absorbance spectra of the resulting CR solutions (in Fig. 6g) show that characteristic peaks at 498 nm and 338 nm almost disappear due to the reduction of azo groups and a new peak at 254 nm comes out assigned to biphenyl formation. Hence, we concluded that a poly-Pd@RCC3 membrane not only separates different dyes, but also catalytically decomposes organic azo dyes. This feature is utilized to self-clean the membrane with the NaBH$_4$ solution in methanol (Fig. 6h). A new poly-Pd@RCC3/PAN membrane has slightly lower methanol permeance than a new poly-RCC3/PAN membrane, since the solvent transport through the cages is partially blocked by the insertion of the Pd nanoclusters. As for the poly-RCC3/PAN membrane, the poly-Pd@RCC3/PAN was cleaned with 100 mL methanol and 20 mL 500 mg/L NaBH$_4$ solution in methanol after the CR dye solution filtration and dye rejection evaluation. The methanol permeance of the poly-RCC3/PAN membrane irreversibly decreased by 13.5%. On the other hand, the methanol permeance of the poly-Pd@RCC3/PAN membrane was recovered after the same cleaning procedure. This was due to the catalytic degradation of the adsorbed CR molecules, recovering the clean membrane surface.

## Discussion

An interfacial polymerization approach is proposed to form a cross-linked network of porous organic cages (POCs) on porous support. The procedure was only possible by increasing the solubility of POCs in water with the addition of an inorganic acid (e.g., HCl) or 1, 2, 3-trifluoroethanol (TFE). The chosen cages were RCC3 and tren, leading to unique structures and properties with well-tailored topologies, and microporosity. Ultrathin selective layers (down to 9.5 nm) were obtained with a high content of cages (70–80 wt%). The low thickness and their intrinsic microporosity endowed membranes with permanent channels for fast solvent permeance. A hyper-cross-linked structure of PCN could be obtained by reacting cages with multi-functionalized sites and TMC. High stability in harsh organic solvent environments was obtained, which is ideal for OSN applications. The PCNs exhibited very high permeance for both polar and nonpolar solvents and high size selectivity for dyes, even superior to most commercial OSN membranes. Molecular simulations provided a vivid presentation of the porous membranes to explain their distinct membrane performance at the molecular level. These simulated results emphasized the elaborated design of the membrane by the introduction of POC to achieve a delicate balance between solvent permeance and size selectivity for an effective targeted molecules

separation. Moreover, a unique membrane with high catalytic activity was fabricated with Pd@RCC3, demonstrating the potential of current membrane production technologies and POCs in multifunctional membrane synthesis. Therefore, this work will inspire the development of cage-based polymeric membranes with well-tailored structures and properties, which could be later extended to photo-, electro- or ionic responsivity, by rational design of molecular building blocks.

## Methods

### Synthesis of amine-functionalized tren-cage

Refer to the reported synthesis protocol[45]. Specifically, 3 g (14.4 mmol) of diphenyl-4,4′-dicarbaldehyde was dissolved in 250 ml acetonitrile in a round-bottom flask. Dropwise over 1 h, a solution of 1.40 g (9.6 mmol) of N1,N1-bis(2-aminoethyl)ethane-1,2-diamine (tren) in 250 ml acetonitrile was added. The reaction was stirred overnight, after which a white precipitate formed, which was filtered, washed with diethyl ether, and dried under vacuum to get 3.10 g, 80% of the tren-imine cage. The material was used directly for the next step as follows, 2 g (2.4 mmol) of the tren-imine cage was suspended in 250 ml methanol, 5 g of solid sodium borohydride (NaBH$_4$, 132.4 mmol) were introduced in portions over a 30 min period. After that, the mixture was stirred for three hours until it was clarified. The solvent was removed under vacuum to dry the remaining solid, which was suspended in 100 ml of dichloromethane (DCM), and extracted with a saturated solution of sodium bicarbonate. The aqueous layer was extracted 2 × 20 ml DCM. The organic layer was dried and evaporated to get a white solid and dried further under vacuum to get 2.4 g, 80% of amine-functionalized tren-cage or tren-cage, which was directly used without further purification. The NMR data for the imine intermediate and the final product was in agreement with the reported literature and was included below. $^1$H NMR (tren-imine cage) (CDCl$_3$, 400 MHz) δ 8.28 (s, 6H), 7.24 (d, J = 4 Hz, 12H), 7.11 (d, J = 4 Hz, 12H), 3.83 (m, 12H),2.82 (m, 12H). $^{13}$C NMR (CDCl$_3$, 125 MHz) 52.6, 57.6, 126.5, 128.5, 135.3, 140.9, 161.7; $^1$H NMR (amine-functionalized tren-cage) (CDCl$_3$, 400 MHz) δ 7.08 (d, J = 4 Hz, 12H), 6.97 (d, J = 4 Hz, 12H), 3.78 (m, 12H), 2.92 (m, 12H), 2.71 (m, 12H). $^{13}$C NMR (CDCl$_3$, 125 MHz) 47.4, 52.4, 54.2, 126.7, 127.9, 138.6.

### Synthesis of amine-functionalized RCC3 cage

For the synthesis of the CC3 cage, refer to a classic method[14]. Specifically, dichloromethane (DCM, 100 mL) was slowly added onto solid 1,3,5-triformylbenzene (5.0 g, 30.86 mmol) without stirring at room temperature. Trifluoroacetic acid (1000 µL) was added directly to this solution as a catalyst for imine bond formation. Finally, a solution of (R, R)−1,2-diaminocyclohexane (5.0 g, 44.64 mmol) in DCM (100 mL) was added. The unmixed reaction was covered and left to stand. Over 5 days, all of the solid triformylbenzene was consumed, and the crystalline product was collected by filtration and washed with pure methanol (5 × 50 mL). The solid was dried at 80 °C in a vacuum oven for the next step of preparation of the amine-functionalized RCC3 cages. Yield: 6.5 g, 83%. $^1$H NMR (CDCl$_3$, 400 MHz): δ 8.16 (s, CH = N, 12H), 7.90 (s, ArH, 12H), 3.34 (m, CHN, 12H), 1.9-1.4 (m, CH$_2$, 48H) ppm. $^{13}$C NMR (CDCl$_3$, 125 MHz): δ 159.12, 136.63, 129.56, 74.61, 33.01, 24.37.

For the synthesis and purification of RCC3, refer to a previous protocol[44]. Specifically, NaBH$_4$ (9 g, 238.5 mmol) was added to a solution of CC3 cage (2 g, 1.76 mmol, 1.0 eq.) in a mixture of chloroform and methanol (1:1 v/v, 400 mL) at 0 °C. The ice-water bath was removed, and the resulting solution was stirred at room temperature for 29 h. The solvent was removed under a reduced vacuum using a rotary evaporator until a gel-like product was obtained. The off-white solid was dispersed into deionized (DI) water (100 mL), extracted with DCM (40 mL), and washed with DI water (3 × 150 mL). The DCM phase was collected, dried with anhydrous Na$_2$SO$_4$, and evaporated under a vacuum using a rotary evaporator. The obtained off-white RCC3 (quant.) was dried at 80 °C under vacuum overnight. To purify the

obtained RCC3, in a 500 mL flask, 1 g of crude RCC3 was dissolved in 100 mL of acetone. The solution was covered and left to stand. Crystals started appearing on the wall of the flask after 30 min. The crystals were collected after 48 h by centrifugation and then dissolved in 100 mL of dichloromethane and methanol mixture (v/v, 1/1) by continuous stirring. About 1 mL of DI water was added to the solution and the mixture was stirred for another 72 h. After the removal of the solvents, the pure RCC3 (600 mg, 60%) was recovered and used for the preparation of the cage aqueous solutions. $^1$H NMR (CDCl$_3$, 400 MHz) δ 7.10 (12H, s), 3.83 (12H, d, J = 14.0 Hz), 3.56 (12H, d, J = 14.1 Hz), 2.18 (12H, d, J = 8.9 Hz), 1.97 (12H, d, J = 12.7 Hz), 1.62 (12H, d, J = 8.4 Hz), 1.17 – 1.12 (12H, m), 0.97 – 0.94 (12H, m); $^{13}$C NMR (CDCl$_3$, 125 MHz): δ 141.38, 125.08, 61.40, 50.75, 31.79, 25.08.

## Synthesis of Pd@RCC3 cage
The synthesis protocol was reported before[43]. In a typical experiment, the 50 mg (0.044 mmol) of RCC3 cage was well dispersed in 10 mL of water and stirred for 30 min. After this, 20 μL of DCM solution containing 0.006 mmol (1.3 mg) of palladium(II) acetate was added using the micro-pipette while vigorous stirring was continued for an additional 3 h. The freshly prepared 2.7 M aq. NaBH$_4$ solution (0.5 mL) was added at once and kept stirred for a further 3 h. The color of the solution turned light brownish solution from light yellowish, which confirms the completion of the reduction process. The obtained aqueous solution was introduced to centrifugation above 15,000 RPM and further washed with water to remove the traces of NaBH$_4$. The synthesized solid was dried in air at 70 °C for 2.5 h and used as it wherever required.

## Preparation of cage aqueous solutions
About 1 g of RCC3 cage was dispersed in DI water (99 ml) by sonication for 10 min. The pH of the above turbid mixture was finely adjusted to around 6.6 by the addition of HCl (0.1 M) aqueous solution. The resulting solution with a concentration of about 1 wt% became transparent and clear, which was ready for membrane preparation after filtration with a syringe filter (0.45 μm of pore size). Similarly, a tren-cage was also used for solution preparation. But a clear solution with a concentration of 0.05 wt% was formulated for membrane preparation. Because the tren-cage in an aqueous solution of the higher concentration was not prepared successfully due to poor solubility.

By the other strategy, a certain tren-cage was dissolved into the TFE-H$_2$O solution with 1,2,3-trifluoroethanol (TFE) to water volume ratio of 1.5, followed by sonication for 5 min. The clear solution with a concentration of 0.3 wt/v% was directly obtained for membrane preparation. The maximum concentration in this solution is about 0.5 wt/v%. Similarly, RCC3 and Pd@RCC3 were used for solution preparation, except the volume ratio of TFE to water was 1. Finally, 0.5 wt/v% solutions were obtained for membrane preparation. The maximum concentration of RCC3 in the solution is about 1 wt/v%.

## Membrane preparation
The protocol of polycage thin-film composite (TFC) membranes by interfacial polymerization was described in Supplementary Fig. 13b. In detail, an asymmetric PAN substrate (GMT GmbH, Rheinfelden, Germany) was fixed with PTFE frames and then wet by 10 mL cage aqueous phase solution for 5 min. After the removal of an aqueous solution, the substrate surface was further wiped with a rubber roller. After re-fixing the frames, trimesoyl chloride (TMC) solution in hexane with a specific concentration was poured on top of the wet substrate. After a certain time, a polycage TFC membrane was rinsed with 60 mL hexane to remove unreacted TMC. The resulting membrane was directly stored in DI water at 4 °C for next use and characterization unless otherwise stated.

Free-standing polycage nanofilms were formed at the free interface between aqueous and organic phases using the same parameters

as those of TFC membranes. For the next characterizations, the formed nanofilms could be deposited on proper substrates according to the procedure in Supplementary Fig. 13a.

## Chemical and thermal characterizations
Attenuated Total Reflection-Fourier transform infrared (ATR-FTIR) spectra was obtained on a Nicolet iS10 spectrometer. Thirty-two scans and a resolution of 4 cm$^{-1}$ were set. Besides, elemental analysis was obtained from X-ray photoelectron spectroscopy (XPS), which was performed on an Axis-Ultra DLD spectrometer using Al Kα radiation (hν = 1486.6 eV). The base pressure was below $3 \times 10^{-9}$ mbar. The C 1s signal of aromatic carbon 284.5 eV was used as a reference to calibrate binding energy data. Detailed calculations could be found in our previous literature[12].

Thermogravimetric analysis (TGA) was implemented on the TA-Q500 with a temperature ramp of 10 °C min$^{-1}$ and a nitrogen flow rate of 10 mL min$^{-1}$.

## Morphological characterizations
Scanning electron microscopy (SEM) images on Magellan were collected in a high-resolution mode with an in-lens detector. A beam voltage of 5 kV, a beam current of 50 pA, and a working distance of 4 mm were set. The samples were cryogenically fractured in liquid nitrogen. They were fixed on the aluminum holders, then covered by 3 nm iridium in a Quorum Q150T. Atomic force microscopy (AFM) images were collected on a Dimension ICON scanning probe microscope, which was operated under tapping mode in the air by using FESPA etched silicon probes (spring constant = 2.8 N m$^{-1}$) with a scan rate of 1 Hz. Wide-angle X-ray diffraction (XRD) patterns were collected on a Bruker D8 Ultra with the scan range from 5 to 90° in a scan speed of 5° min$^{-1}$.

The free-standing polycage nanofilms were directly collected on the PELCO® 200 mesh copper transmission electron microscope (TEM) grids (Ted Pella Inc., CA, USA) and stored in the dry cabinet prior to visualization. The samples were imaged on a Titan ST high-resolution TEM (HR-TEM) (FEI, OR, USA) at 300 kV accelerating voltage, extraction 4000 V, and spot size 3; all the micrographs were made with the use of 50 μm objective aperture. The solution of Pd@RCC3, 3 μL of 0.5 wt/v% of Pd@RCC3 in TFE-H$_2$O solution was collected on the glow-discharged C-Flat Standard carbon Multi-A TEM grid (Protochips, NC, USA), the excess of the solution was blotted with a filter paper and dried for 24 h prior to visualization. The sample was visualized with Titan ST HR-TEM (FEI Company, OR, USA) at 300 kV accelerating voltage, extraction 4000 V, and spot size 3, 50 μm objective aperture was used.

## Pure solvents filtration and organic solvent nanofiltration test
Pure solvents filtration and organic solvent nanofiltration (OSN) test were carried out using a commercial dead-end cell (Sterlitech, HP4750). The feed volume was 200 mL and the effective membrane area ($A$) was 13.8 cm$^2$. For pure solvent filtration, each membrane sample was stabilized for 1 h for each solvent or each applied pressure. The applied pressure was set at 0.5–10 bar, depending on the permeance of each solvent. The weight of the permeate was auto-recorded every 2 min to calculate pure organic solvent permeance. To study the influence of the applied pressure on the permeate flux, applied pressures varied from 1 to 10 bar. For the OSN test, a series of single dye (Rose Bengal, RB, 974 g mol$^{-1}$; Congo Red, CR, 697 g mol$^{-1}$; Acid Fuchsin, AF, 586 g mol$^{-1}$; Orange G Sodium, OGS, 452 g mol$^{-1}$; Methyl Orange, MO, 327 g mol$^{-1}$; and Methyl Red, MR, 269 g mol$^{-1}$) solutions in methanol were used to determine the membrane selectivity. The solution concentration was 20 ppm and the applied pressure was 1 bar unless otherwise stated. Permeate was collected when a steady permeate was achieved. All experiments were conducted three times in parallel at room temperature.

The solvent flux ($J$) is calculated by Eq. 1, where $\Delta w$ is the permeate weight within a time $\Delta t$, A is the effective membrane area, and $\rho$ is the permeate density.

$$J = \Delta w / \rho A \Delta t \tag{1}$$

And permeance is calculated by Eq. 2, where $\Delta P$ is the applied pressure.

$$P = J / \Delta P \tag{2}$$

Rejection for a single dye was calculated using Eq. 3, where $C_f$ and $C_p$ are the concentration of feed solution and permeate, determined by NanoDrop UV-vis spectrophotometer, using quartz cuvettes.

$$R = \left(1 - C_p / C_f\right) \times 100\% \tag{3}$$

### Catalytic test

Poly-Pd@RCC3 nanofilms samples were washed with methanol many times in a glass with a gentle stirring and further dried under vacuum at 80 °C for 24 h before the catalytic test. A procedure for the catalytic decomposition of organic dyes by using poly-Pd@RCC3 membranes and Pd@RCC3 was performed[43]. Typically, a Congo red (CR) solution (4 ml, 0.05 mmol L$^{-1}$) in methanol was mixed with 6 mg or more NaBH$_4$ at room temperature in a quartz cell. Then, a mixture of 10 μl of Pd@RCC3 (around 0.000011 mmol Pd) or some pieces of poly-Pd@RCC3 nanofilms was added to the solution and the catalytic reactions were monitored using UV-Vis.

The feature of catalytic decomposition of organic dyes was utilized to self-clean poly-Pd@RCC3 membranes with the help of the NaBH$_4$ solution in methanol. Specifically, a fresh poly-RCC3 membrane (as a reference) or poly-Pd@RCC3 membrane was evaluated with pure methanol filtration and CR dye rejection in a methanol medium. Permeate was collected when a steady permeate was achieved. And then, membranes were cleaned with 100 mL methanol and 20 mL 500 mg/L NaBH$_4$ solution in methanol under vigorous stirring. Repeat the pure methanol filtration and CR dye rejection in methanol medium again by using cleaned membranes as a comparison.

## Data availability

All data that support the findings of this study are available within the paper and Supplementary Information files or from the corresponding author upon request.

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

## Acknowledgements

The authors thank King Abdullah University of Science and Technology (KAUST) for the financial support.

## Author contributions

X.L., M.B., N.M.K., and S.P.N. conceived the project. X.L. designed the experiments. X.L., W.L., V.S., and M.G. carried out the materials synthesis. X.L., R.G., B.A.M., S.H., and S.A. performed materials characterization. X.L. carried out pure solvent filtration, organic solvent nanofiltration, and catalytic test. X.L. and S.P.N. analyzed the data. X.L. and W.L. conducted molecular modeling. X.L., S.P.N., and N.M.K. wrote the manuscript. All authors discussed the results and commented on the manuscript.

## Competing interests

The authors declare no competing interests.
