## [Peer Review File · Nature Communications]

Polycage membranes for precise molecular separation and catalysisReviewers' Comments:

Reviewer #1:

Remarks to the Author:

In this article, authors represented two novel membranes prepared by interfacial polymerization between amino-polycages and TMC, and the membrane thicknesses are very thin, and they have a high permeance. In the two cages, catalyst has been filled, and the membranes have been endowed a dual function of the separation and catalysis. But I don't think the article can be accepted, for many questions are unclear in it. Some comments are listed as follow.

The author has given a schematic diagram of transferring the free-standing film to the support layer in Supplementary Fig. 13, but I think the transfer is a very delicate operation, and is it easy to be applied in actual production? How is the success rate of freestanding film transfer?

Why does the thickness of the PCN1-TFE film first increase and then decrease over the concentration of TMC? In my opinion, the thickness of membrane could keep constant with the increase of TMC concentration

There is a strong peak at 837cm⁻¹ for PCN1-TFE film, and authors claimed that the result demonstrate the existence of a side reaction between TMC and TFE, and I can't sure if it's a residual TFE or a byproduct, and I suggested that authors give a clear answer, and how to clean the free-standing PCN1-TFE film?

Authors thought that the much lower concentration of tren-cage in an aqueous solution than RCC3 amino-functionalized cages greatly reduces the diffusion rate, and does just the concentration affect the diffusion rate?

The methanol and water permeances of PCN1-acid membranes decrease with the increase of reaction time, and authors thought that the trend can be correlated with the thickness, and I don't sure if the density and cross-linking degree of membranes will have different with the change of reaction time, and they will affect the permeance.

What are the pore sizes of tren-cage and RCC3? And I think the different rejection for dyes should be explained in conjunction with their pore sizes.

One of reason for the different rejection of dyes is the more effective crosslinking for the membrane prepared by tren-cage, but in membrane chemical characterization section, is the cross-linking degree of RCC3-based membrane higher?

Several organic solvents permeation has been reported in this article, but the change of the solvent permeance has just been analyzed according to the viscosity. The permeance of solvent is very complex, and is concerned with solvent size, viscosity, and the interaction with membrane, so I don't think the analysis is very reasonable.

Reviewer #2:

Remarks to the Author:

The manuscript reports a relatively new class of polyamide (PA) membranes fabricated using interfacial polymerization. The key novelty is that, instead of using amine monomers (such as PEI, PIP, or MDP), the team engineered amino-functionalized organic cages (tren-cage and RCC3) to react with acid chloride. The authors also impregnated the cages with Pd catalysts to impart the ability of catalytic self-cleaning.

From a chemistry and material design perspective, the work is interesting and novel. From a performance perspective, the novelty is less clear. While the authors performed comprehensive characterizations and performance testing in organic solvent nano-filtration (OSN), it is unclear in what way these membranes outperform existing membranes in terms of separation. What is the motivation behind designing such new membranes? What technical challenges the new membranes are intended to overcome?

It may be that the value proposition is that the cage-based PA can enable the imbedding of Pd

catalysts to generate a catalytic membrane. But then what is the exact purpose of a catalytic membrane? I was expecting that the authors show some comparative fouling experiments to demonstrate that with catalytic membrane foulants are actively destroyed and the performance is much more stable, which does not seem to be the case. The figure 6h tries to convey is unclear even after reading the manuscript. Additionally, what's the advantage putting catalyst in the cages vs. the many existing approaches of making catalytic membranes via surface coating or embedding.

What I found scientifically interesting is the "dip" in rejection for the second molecular weight in figure 4b, d and f, which appears to be systematic and not an experimental error. Have the authors tried to perform such characterization on a normal PA membrane and observed similar behavior? If this is unique to cage-based PA membrane, the science behind it would be quite interesting (although not necessarily useful).

Overall, I feel that the authors need improve the manuscript by conveying a clearer message of rationales and merits of reported membranes. In its current form, it just seems no more than "just another way of making PA membrane". Whether the new chemistry is sufficient for publication in this journal is a question for the editor.

Reviewer #3:

Remarks to the Author:

This is an interesting paper that offers insight into the use of polycages as a new class of membrane filler material in mixed matrix membranes. The membranes utilize a conventional polyamide-type TFC membrane as a platform to hold the polycages. This type of work harkens back to the early days of thin film nanocomposite membranes which first introduced zeolites into the polyamide matrix in an attempt to improve permeance and selectivity. This approach appears to be similar to is more likely to have broader applications (such as organic solvent separations) as long as the long term viability of the membrane is retained within a relevant environment.

While the performance of the resulting membranes is good, the paper lacks context for the work. Why specifically is this work important. Simply saying that molecular separations is needed is vague and not telling the entire story.

I also would have liked to have seen some degree of resiliency testing. How will the membrane hold up longer term to these solvent-laden environments? The rejection data also has limited fidelity. There needs to be a finer molecular probe that provides higher resolution of selectivity (such as in Figure 4B).

The discussion section lacks any discussion. It is written like a conclusion that summarizes the paper rather than discusses the scientific and technological implications of the work.

In all, this is a very interesting membrane science paper and would fit well within a membrane journal.

As a small detail, I was unsure why a PAN substrate was used rather than a more traditional polysulfone support?

Response to the Editor and Reviewers

We submit here the revised manuscript addressing all points raised by the reviewers detailing our responses one by one.

Please note that we changed the membrane codes in the revised version. PCN1 became tren; PCN2 became RCC3. This facilitates the identification of the membranes directly linking them to the constituting cages.

Response to Review #1

Reviewer #1 (Remarks to the Author):

[General Comment] In this article, authors represented two novel membranes prepared by interfacial polymerization between amino-polycages and TMC, and the membrane thicknesses are very thin, and they have a high permeance. In the two cages, catalyst has been filled, and the membranes have been endowed a dual function of the separation and catalysis. But I don't think the article can be accepted, for many questions are unclear in it. Some comments are listed as follow.

Response: Thanks very much for your comments. We carefully read and addressed them all in detail here. Your suggestions and questions will certainly help to improve the manuscript.

[Comment 1] The author has given a schematic diagram of transferring the free-standing film to the support layer in Supplementary Fig. 13, but I think the transfer is a very delicate operation, and is it easy to be applied in actual production? How is the success rate of freestanding film transfer?

Response: Thank you for pointing this out. We agree with you that the transfer is a delicate operation, and it is not practical to be applied in actual production. The purpose of the procedure described in Fig. 13 is the sample preparation for characterization (e. g. thickness, roughness, element analysis) by atomic force microscopy (AFM) and X-ray photoelectron spectroscopy (XPS), without the interference of an asymmetric porous support. For the membrane performance experiments, polycage thin-film composite (TFC) membranes were prepared on PAN porous supports. This reflects the procedure that

would be applied in a membrane production. The kind of support would be the same. This support would be immersed in an aqueous (or mixed solvent) solution containing the first monomer, the excess would be removed, and the system would be brought in contact with the organic solution with the second monomer. In the production, instead of pieces of support, we would use a continuous machine with roles of PAN support. A machine like that is available in our lab. We now extended Fig. 13 to represent the different applied procedures also on PAN support.

[Comment 2] Why does the thickness of the PCN1-TFE film first increase and then decrease over the concentration of TMC? In my opinion, the thickness of membrane could keep constant with the increase of TMC concentration.

Response: Thanks for your question about the thickness of the tren-TFE film. We agree with your opinion in the case that the membrane is produced from TMC and MPD at the water/alkane interface by classic interfacial polymerization. As previously demonstrated (Langmuir 2003, 19, 11, 4791–4797), the thickness of this incipient film forms during the initial period. A tight polymer layer is quickly formed and the monomer interdiffusion is then inhibited. Because the solubility of n-hexane in water is 0.0021v/v%, and the solubility of water in n-hexane is 0.0065v/v% (Talanta 1968, 15(11): 1281-1286), the reaction takes place at the organic phase side. Accordingly, the amine concentration determines the amine diffusion rate from aqueous phase to organic phase. The thickness of the membrane could increase with the increase of TMC concentration from extreme low value but becomes constant at a certain concentration.

But in this manuscript, the initial reaction zone at the TFC-H₂O/hexane interface is different. Firstly, the solubility of n-hexane in 2,2,2-trifluoroethanol is 2.8v/v%, and the solubility of 2,2,2-trifluoroethanol in hexane is around 1.3v/v% (*Russian Journal of Physical Chemistry A*, 2009, 83, 1966–1971; *Journal of Chromatography A*, 2017, 1527, 18-32.). Compared with the H₂O/hexane interface, the initial reaction zone is wider. Furthermore, the surface tension of 2,2,2-trifluoroethanol is 21.1 mN m⁻¹, much lower than that of water (72.75 mN m⁻¹). The interface is much more diffuse and does not stop the monomer transport. In the beginning, the increase of TMC concentration from 0.005 wt/v% to 0.025 wt/v% promotes the diffusion of TMC from the hexane phase to the H₂O-TFE

phase and leads to thicker films. When the TMC concentration continues to increase up to 0.1 wt/v%, the ratio of TMC to amine-functionalized cage increases and a thinner film is formed. The film thickness is then limited by the incipient film itself. An even higher concentration (0.15 wt/v%) doesn't further lead to a thinner film.

We added these comments to the manuscript.

[Comment 3] There is a strong peak at 837cm⁻¹ for PCN1-TFE film, and authors claimed that the result demonstrate the existence of a side reaction between TMC and TFE, and I can't sure if it's a residual TFE or a byproduct, and I suggested that authors give a clear answer, and how to clean the free-standing PCN1-TFE film?

Response: Thanks for your question. During the synthesis of a free-standing tren-TFE film, a possible side product could be formed in the reaction between TFC/water and TMC. Small side products would be trimesic acid and tris[2,2,2-trifluoroethyl]-1,3,5-benzenetricarboxylate, which are soluble in methanol. The free-standing tren-TFE film sample can be easily washed with methanol many times in a glass via with a gentle stirring, and further dried under vacuum at 80 °C for 24 h.

[Comment 4] Authors thought that the much lower concentration of tren-cage in an aqueous solution than RCC3 amino-functionalized cages greatly reduces the diffusion rate, and does just the concentration affect the diffusion rate?

Response: Thank you for pointing this out. Besides the amine-functionalized cages concentrations in the aqueous solution, the difference in degree of protonation affects the reactivity and the solubility in the apolar phase.

[Comment 5] The methanol and water permeances of PCN1-acid membranes decrease with the increase of reaction time, and authors thought that the trend can be correlated with the thickness, and I don't sure if the density and cross-linking degree of membranes will have different with the change of reaction time, and they will affect the permeance.

Response: Thanks very much for your comments. The reviewer is completely right to consider that not only thickness but also the crosslinking degree and the layer density would affect membrane performance. For the same thickness, a polyamide membrane

with a higher crosslinking degree should have a higher density and a lower permeance. To evaluate the crosslinking degree of tren-acid membranes, we measured by XPS the content and kind of bond of N in different membranes, comparing tren-acid (reaction time 30min) and tren-acid (reaction time 10 s), as shown below. Based on the N1s spectra in both samples, the percentage of the reacted -NH- group per tren-cage molecule could be calculated according to the ratio of the peak of N-C=O (399.7 eV) to the whole signal. 40 % and 34% of the -NH- groups were consumed during the interfacial polymerization and formation of the tren-acid (reaction time 30min) and tren-acid (reaction time 10 s) layers, respectively. Therefore, we conclude that a shorter reaction time leads to a membrane with lower crosslinking degree, and this affects the permeance as well. We added this aspect to the discussion.

N1s spectra of tren-acid with reaction time (a) 30 min and (b) 10 s.

[Comment 6] What are the pore sizes of tren-cage and RCC3? And I think the different rejection for dyes should be explained in conjunction with their pore sizes.

One of reason for the different rejection of dyes is the more effective crosslinking for the membrane prepared by tren-cage, but in membrane chemical characterization section, is the cross-linking degree of RCC3-based membrane higher?

Response: We agree that the rejection of different dyes should be explained in conjunction with the selective layer “pore” (of transport path) size, besides the cross-

linking degree. According to the structural analysis of CC3 cage and FT-RCC3 cage (*CrystEngComm*, 2017, 19, 4933–4941) and molecular calculation by Mercury, the transport channels (“pores”) in RCC3 cages are expected to be larger than 0.7 nm. In tren-cages membranes, they are smaller than 0.5 nm. We added a **Supplementary Fig. 28** in the revised manuscript as shown below to give a clearer information.

Supplementary Figure 28. Possible transport channels through cages (described by green ball). **a**, Cavity size of 15 Å; **b**, window size of 12.4 Å for RCC3-cage; **c**, cavity size of 4.7Å; **d,e** window size of 3.9 Å and 6.0 Å for tren-cage. Scale bar is 10 Å.

As the reviewer mentioned, in the discussion on the membrane chemical characterization, we mention that the crosslinking degree of RCC3-cage membrane is higher than that of the tren-cage one. Compared with common smaller amines used for interfacial polymerization, the molecular topology of large cages and their packing types (*Nature*, 2022, 609, 58–64) might affect the formation of effective pores/void between cages in the final selective layers. In this work, one-dimensional tren-cages with biphenyls might pack tighter than three-dimensional RCC3 cages, due to additional pi-pi interaction between two biphenyls in adjacent cages. This is another reason to explain why the membrane prepared with tren-cage gave better rejection of different dyes. We added two **Supplementary video 1 and 2** to illustrate the molecular topology.

The original manuscript was revised after adding Supplementary Fig. 28.

[Comment 7] Several organic solvents permeation has been reported in this article, but the change of the solvent permeance has just been analyzed according to the viscosity. The permeance of solvent is very complex, and is concerned with solvent size, viscosity, and the interaction with membrane, so I don't think the analysis is very reasonable.

Response: Thanks for your comment. We agree that the permeance of solvent is complex, and for nanofiltration membranes it may be a combination of solvent size, viscosity and the interaction with the membrane material. A simple way to evaluate the extent of the contribution of different factors to the transport in OSN membranes has been adopted by the Livingston's group and by others (see for instance *Science*, 2015, 348, 1347-1351). A linearity of permeance vs $1/\eta$ indicates the viscous nature of the solvents' flow through the membranes, following the classical Hagen-Poiseuille equation. If other factors like the thermodynamic interaction between solvent and membrane material would be relevant, a more complex plot would give a more linear behavior following the equation below

$$P_{s,i} = K_i \left(\frac{\delta_{p,s}}{\eta \cdot d_{m,s}} \right)$$

where $P_{s,i}$ ($\text{L m}^{-2} \text{h}^{-1} \text{bar}^{-1}$) is permeance of solvent s , K_i is a proportionality constant for nanofilm i ($\text{m}^3 \text{Pa}^{-0.5}$), $\delta_{p,s}$ is the solubility parameter ($\text{Pa}^{0.5}$), η_s is the solvent viscosity ($\text{Pa}\cdot\text{s}$), and $d_{m,s}$ is the molar diameter of the solvent s (m).

We plotted in different ways the solvent permeance through our polycage TFC membranes, as seen in the **Supplementary Fig. 32**, reported as below. However, it seems that the plot with the inverse of viscosity describes the permeance of solvent through this ultrathin porous polycage TFC membranes very well. Similarly, this behavior of solvent transport was studied in nanoporous graphene and covalent organic framework membranes (*Nature Materials* 2017, 16, 1198–1202; *Journal American Chemistry Society*, 2018, 140, 43, 14342–14349).

Supplementary Figure 32. Solvents permeance vs viscosity and inverse viscosity through **a, b, c**, tren-acid (reaction time 10 s) membrane, **d, e, f**, tren-TFE-0.07 membrane, **g, h, I**, RCC3-acid -(reaction time 5 min) membrane and **j, k, l**, RCC3-TFE-0.005.

The main text was revised after the inclusion of the *Supplementary Fig. 32*:

Response to Review #2

Reviewer #2 (Remarks to the Author):

[General Comment] The manuscript reports a relatively new class of polyamide (PA) membranes fabricated using interfacial polymerization. The key novelty is that, instead of using amine monomers (such as PEI, PIP, or MDP), the team engineered amino-functionalized organic cages (tren-cage and RCC3) to react with acid chloride. The authors also impregnated the cages with Pd catalysts to impart the ability of catalytic self-cleaning.

Response: Thanks very much for your comments. We have carefully read your comments and addressed them one by one.

[Comment 1] From a chemistry and material design perspective, the work is interesting and novel. From a performance perspective, the novelty is less clear. While the authors performed comprehensive characterizations and performance testing in organic solvent nano-filtration (OSN), it is unclear in what way these membranes outperform existing membranes in terms of separation. What is the motivation behind designing such new membranes? What technical challenges the new membranes are intended to overcome?

Response: Thank you very much for agreeing with us to the novelty and interest of this work from a chemistry and materials design perspective. Cages have been used before for membrane preparation, but it is the first time that they are fully crosslinked and applied with the approach reported in this paper. We believe that in this form the advantage of the cages cavities and channels will be predominant. The development of new structures in the form of cages and similar building blocks is a growing fast and we demonstrate what we consider the best method to integrate them into membrane science in a scalable way. From a performance perspective, we agree that the membranes might not have the record of most permeable or most selective in OSN. We demonstrate polycage membrane which are highly effective for the precise separation of molecules with similar sizes in the range of 600 g/mol, compared with existing membranes. Due to the nature of

the method that we propose, it could be easily translated in machines already used in the industry.

The motivation behind designing such new membranes is to develop polymer membranes with high selectivity, to distinguish between molecules of similar sizes and charges. Material science has developed a variety of structures that could be excellent tools for the separation of small molecules. Examples are porous organic cages. But engineering cages in a form that could be used for effective molecular separations in small and large scales is as challenging as the cage design itself.

Besides high selectivity, another technical challenge the new membranes are intended to overcome is to integrate other functions into membranes. Here, the catalytic and self-cleaning properties are added. But it's conceivable that other functions such as photo-responsibility and temperature-responsivity are also obtained through regular chemistry in porous organic cages by our proposed method for membrane preparation.

We improved the introduction to highlight the advantages.

[Comment 2] It may be that the value proposition is that the cage-based PA can enable the imbedding of Pd catalysts to generate a catalytic membrane. But then what is the exact purpose of a catalytic membrane? I was expecting that the authors show some comparative fouling experiments to demonstrate the with catalytic membrane foulants are actively destroyed and the performance is much more stable, which does not seem to be the case. The figure 6h tries to convey is unclear even after reading the manuscript. Additionally, what's the advantage putting catalyst in the cages vs. the many existing approaches of making catalytic membranes via surface coating or embedding.

Response: Thank you very much. We used one catalytic reaction to demonstrate that the membrane prepared with this approach could have a double function. We followed the example of Yang et al. *Nature Catalysis* 2018, 1, 214 (Ref 42) and demonstrated in Figure 6 that the catalyst activity is retained in the membrane with the crosslinked cages.

The interesting about this approach is the possibility of having the catalyst highly dispersed in nanoclusters of controlled size integrated in the membrane. The results demonstrated in this manuscript are a proof of concept for catalytic activities combined

with separation and selectivity. However, it was not the purpose to have the membrane tested for regular fouling.

[Comment 3] What I found scientifically interesting is the “dip” in rejection for the second molecular weight in figure 4b, d and f, which appears to be systematic and not an experimental error. Have the authors tried to perform such characterization on a normal PA membrane and observed similar behavior? If this is unique to cage-based PA membrane, the science behind it would be quite interesting (although not necessarily useful).

Response: It’s a nice suggestion. The “dip” intrigues us too. To investigate it further, following the reviewer suggestion, a classical (normal) polyamide membrane was prepared on a PAN support as control under comparable conditions: 2 wt% MPD aqueous solution and 0.1 wt/v% TMC solution in hexane, reaction time of 1 min, followed by 5 min treating at 80 °C. The rejection is above 94% for all tested dyes and there is no “dip” (see below). The free space for permeation in the control membrane is random. When cages are part of the membrane, the paths have a well-defined size. We believe that the membranes working in the range of molecular weight below 500 g mol⁻¹ with the structures investigated here can distinguish not only size but molecular shape. Please see the structures of the dyes. While Methyl Orange (MO) has a molecular weight 327 g mol⁻¹, larger than Methyl Red (MR, 269 g mol⁻¹), MR is bulkier than MO. MO molecules can more easily permeate through the cages. We added a sentence on this effect to the main text.

Methyl Red (MR, 269 g mol⁻¹)

Methyl Orange (MO, 327 g mol⁻¹)

Orange G Sodium (OGS, 452 g mol⁻¹)

[Comment 4] Overall, I feel that the authors need improve the manuscript by conveying a clearer message of rationales and merits of reported membranes. In its current form, it just seems no more than “just another way of making PA membrane”. Whether the new chemistry is sufficient for publication in this journal is a question for the editor.

Response: Thanks for your comment. We are highlighting even more what are the potential advantages of this class of materials.

Response to Review #3

Reviewer #3 (Remarks to the Author):

[General Comment] This is an interesting paper that offers insight into the use of polycages as a new class of membrane filler material in mixed matrix membranes. The membranes utilize a conventional polyamide-type TFC membrane as a platform to hold the polycages. This type of work harkens back to the early days of thin film nanocomposite membranes which first introduced zeolites into the polyamide matrix in an attempt to improve permeance and selectivity. This approach appears to be similar to is more likely to have broader applications (such as organic solvent separations) as long as the long term viability of the membrane is retained within a relevant environment.

Response: Thanks very much for your interest and recognition in polycages for membrane community. We have carefully read your comments that help us improve this manuscript.

[Comment 1] While the performance of the resulting membranes is good, the paper lacks context for the work. Why specifically is this work important? Simply saying that molecular separations is needed is vague and not telling the entire story.

Response: Thank you. We are emphasizing the potential advantage in the introduction.

[Comment 2] I also would have liked to have seen some degree of resiliency testing. How will the membrane hold up longer term to these solvent-laden environments? The rejection data also has limited fidelity. There needs to be a finer molecular probe that provides higher resolution of selectivity (such as in Figure 4B).

Response: Thanks very much for your nice suggestions.

Firstly, it's demonstrated that polycage films were stable without dissolution after immersion in various organic solvents for 7 days (Supplementary Fig. 34a). Long term pure solvent filtration of more than 72 h was conducted (Supplementary Fig. 34b), and long-term separation of dye solution (acidic fusion) with high concentration (20 ppm) has been obtained after pure solvent filtration as well. Almost full rejection (99.9%) was observed (Supplementary Fig. 34c), without any reduction. The obtained result (Supplementary Figure 34) has been added to supplementary information, as suggested.

Supplementary Figure 34. **a**, tren-acid nanofilms after immersion in various organic solvents for 7 days. **b**, The long-term stability of tren-acid -(reaction time 1 min) membrane and **c**, acid fusion solution in methanol (20 ppm) for more than 3days.

Secondly, as you suggested, we tested a series of finer molecular probes, including dyes mixtures and complex phenolic compound mixtures in mixture based on the tren-acid (reaction time 10s) TFC membrane. To provide more information on the high resolution of selectivity, we investigated two mixtures comprising different molecular sizes, charges

and rigidity. The obtained Supplementary Figure 31 was added to supplementary information, and the main text of the manuscript was revised accordingly.

Supplementary Figure 31. Molecular separation performance in organic mixtures with tren-acid-(reaction time 10 s) membranes on PAN support. **a**, Markers with negative (black line) and neutral (red line) charge in methanol, and 10 μ M for each composition. Inset, Photos of feed, permeate and retentate solutions within testing. **b**, Phenolic mixture in methanol and 10 μ M for each composition. Nanofiltration was conducted in dead-end filtration system at 20 °C under 5 bar. Rejection versus molecular weight of compounds: Coomassie brilliant blue R 250 (BBR, 826 g mol⁻¹), Orange G Sodium (OGS, 452 g mol⁻¹), Methyl Orange (MO, 327 g mol⁻¹), sodium 3,5-dichloro-2-hydroxybenzenesulfonate (DHBS, 260 g mol⁻¹); Vitamin B12 (VB12, 1355 g mol⁻¹), bromothymol Blue (BrB, 624 g mol⁻¹), Coumarin 6 (C6, 350 g mol⁻¹); 2,2-biphenol (186.2 g mol⁻¹), 2,6-di-tert-butyl-4-methylphenol (220.4 g mol⁻¹), bisphenol A (228.3 g mol⁻¹), 4,4' isopropylidenebis (284.4 g mol⁻¹), (R)-(+)-1,1'-Bi-2-naphthol (286.3 g mol⁻¹), 4,4'-(1 phenylethylidene)bisphenol (290.4 g mol⁻¹), 3,3,3',3'-tetramethyl-1,1'-spirobiindane-5,5',6,6'-tetraol (340.4 g mol⁻¹). Composition analysis was conducted with HPLC system.

[Comment 3] The discussion section lacks any discussion. It is written like a conclusion that summarizes the paper rather than discuss the scientific and technological implications of the work.

Response: The discussion was in great part addressed in Section 2. We named the final paragraphs now “Final Discussion”.

[Comment 4] In all, this is a very interesting membrane science paper and would fit well within a membrane journal.

Response: Thanks very much for your comment. We believe the manuscript will be of general interest to the readership, with a focus on porous organic cages, engineered into scalable membranes for molecular separation.

[Comment 5] As a small detail, I was unsure why a PAN substrate was used rather than a more traditional polysulfone support?

Response: Polysulfone is typically used for membranes applied for desalination and therefore in aqueous medium. PAN is industrially used for organic solvent nanofiltration and gas separation. A PAN support is more stable in harsh organic solvents, such as acetone and tetrahydrofuran.

Reviewers' Comments:

Reviewer #1:

Remarks to the Author:

Authors answered all questions very well, and updated their manuscript based on reviewers' comments, I think it can be accepted now.

Reviewer #2:

Remarks to the Author:

I feel the authors have adequately addressed my comments. The work is more focused on new chemistry and material design and it's acceptable the new membrane does not necessarily yield superior performance vs. the state of the art. Acceptance is recommended.

Reviewer #3:

Remarks to the Author:

Thank you for the opportunity to review this manuscript again. The authors addressed some of the more technical comments from the reviewers, but they missed an opportunity to be more persuasive as to the value of this work from a broader perspective, particularly with respect to the state of the art membranes out there for OSN. I reviewed the introduction again and, though I don't know what was actually changed, it lacks a compelling argument for why this work is necessary, particularly for a nascent field like OSN.

The authors seem to struggle to talk about the novelty of this work as it relates to actual impact. Reviewer #2 called this out in their comment #1 regarding membrane performance. The authors simply responded that cages have never been used before. That's not good enough to warrant publication in a general scientific journal. There must be new science included, and simply using a new material for membrane does not constitute that. I had hoped the author would provide a more persuasive argument.

Reviewer # 2 continues to critique the paper regarding the catalytic properties of the membrane which are claimed to offer cleaning benefits, but no fouling or cleaning data was include. The response was simply that the purpose of the paper was not to show that. Then I would ask why bother talking about it? Just because it was described in another paper as having catalytic properties, you should probably show those properties in this paper. Again, this was a poor response to a reasonable critique.

Catalytic OSN membranes don't seem to make much sense unless specific fouling situations exist for certain industrial processes. Most of the fouling work in the literature is limited to the water treatment space. In general, I worry that trying to make a membrane do too many things at once just makes it do all of those things poorly. The authors need a compelling argument for combining the two together. I don't feel that this paper has that argument.

The authors claim in their response that they are highlighting the potential advantages of this membrane, but I don't see an annotated version of the manuscript or the changes described explicitly in the author response. That makes it hard to assess what changes were actually made.

Again in Reviewer 3, comment 1, the reviewer asks for context. The authors just say to refer to the new introduction but fail to explain what was changed.

The authors seemed to miss the point of Reviewers 3 comment 4. This was meant to suggest that the paper belongs in a membrane journal, and not Nature Communications. I'm not convinced that the

authors have made a compelling argument that it belongs in this specific journal. It definitely should be published somewhere, though, but I continue to believe a membrane or separations journal is more appropriate.

Dear Reviewers

We submit here the revised manuscript addressing all points raised by the reviewers detailing our responses one by one. Thank you for all your comments and for the chance of having the final improvements. Please see our detailed responses below.

Response to Review #1

Reviewer #1 (Remarks to the Author):

[General Comment] Authors answered all questions very well, and updated their manuscript based on reviewers' comments, I think it can be accepted now.

Response: Thanks very much for your previous comments to improve our manuscript and for your agreement.

Response to Review #2

Reviewer #2 (Remarks to the Author):

[General Comment] I feel the authors have adequately addressed my comments. The work is more focused on new chemistry and material design and it's acceptable the new membrane does not necessarily yield superior performance vs. the state of the art. Acceptance is recommended.

Response: Thanks very much for your previous comments to improve our manuscript and for your positive recommendation.

Response to Review #3

Reviewer #3 (Remarks to the Author):

[Comment 1] Thank you for the opportunity to review this manuscript again. The authors addressed some of the more technical comments from the reviewers, but they missed an opportunity to be more persuasive as to the value of this work from a broader perspective, particularly with respect to the state of the art membranes out there for OSN. I reviewed the introduction again and, though I don't know what was actually changed, it lacks a compelling argument for why this work is necessary, particularly for a nascent field like OSN.

[Comment 2] The authors seem to struggle to talk about the novelty of this work as it relates to actual impact. Reviewer #2 called this out in their comment #1 regarding membrane performance. The authors simply responded that cages have never been used before. That's not good enough to warrant publication in a general scientific journal. There must be new science included, and simply using a new material for membrane does not constitute that. I had hoped the author would provide a more persuasive argument.

Response to comments 1 and 2: Thanks very much for all your comments. Please see our answer to comment 4 for details on what we changed in the previous revision and now. Reviewers 1 and 2 are now satisfied.

Our main aim in this manuscript is to explore cages as potential pre-formed nanopores or nanochannels to obtain membranes selective for small molecules. We have before explored the use of macrocycles with the same objective. Our expectation is that the transport and interaction through cages would promote even more selectivity, and we started this work years ago. In other words, expect to profit from the highly defined 3D structures of the cages as building blocks, as potentially advantageous in comparison to the rather 2D units of macrocycles. The use of cages has additional challenges of solubility. A few recent manuscripts have now been published using cages for membranes. Those who are the closest to our approach are the following. The first example is the interesting work of He et al., Nature Mater. 2022, 21, 463, where cages are dynamically synthesized and crystallized on an interface and deposited on a porous support. Another example has been recently published by Zhao et al. J. Membr. Sci. 2022, 664, 121081. They use a RCC3 cage combined with piperazine as monomers and tested for filtration of salts and dyes aqueous solutions. This example had not been added before and is now in the new version.

We consider it essential to have (1) full use of the cage properties, (2) fabricate membranes with stability under operating conditions and (3) use scalable methods preferentially translated into industrial machines. The dynamic synthesis/crystallization method is highly interesting from the fundamental point of view but would not provide the needed stability and could be hardly reproduced in machines in the way interfacial polymerization membranes are currently produced, since reaction times of several hours

are required. In the second example, the cages are mixed with amines in the aqueous phase for interfacial polymerization. The disadvantage of this approach is the dilution of what the cage could provide. In our work, we succeeded to have the selective layer based on a fully crosslinked cage network. In this way, we force the permeant to pass through the cages by reducing any alternative network pass, which would otherwise be formed if we added other amines. The method could be performed in a continuous machine as used in industrial processes since the reaction is relatively fast (seconds or minutes instead of hours). To succeed we had to overcome a critical issue, the solubility of the cages in the aqueous phase. The way to do it is an essential innovation. Another important point is that we compare cages with different segments to demonstrate how the cage intrinsic properties influence the future design of membranes. Finally, we propose how to combine a catalytic function with highly distributed (non-aggregated) catalysts with selective permeation.

We added or highlighted the above arguments to the introduction to make the innovation clearer.

[Comment 3] Reviewer # 2 continues to critique the paper regarding the catalytic properties of the membrane which are claimed to offer cleaning benefits, but no fouling or cleaning data was included. The response was simply that the purpose of the paper was not to show that. Then I would ask why bother talking about it? Just because it was described in another paper as having catalytic properties, you should probably show those properties in this paper. Again, this was a poor response to a reasonable critique.

Catalytic OSN membranes don't seem to make much sense unless specific fouling situations exist for certain industrial processes. Most of the fouling work in the literature is limited to the water treatment space. In general, I worry that trying to make a membrane do too many things at once just makes it do all of those things poorly. The authors need a compelling argument for combining the two together. I don't feel that this paper has that argument.

Response: As described above, the most important contribution of this paper is the preparation of fully crosslinked cage membranes by a scalable method enabled by overcoming critical issues related to cage solubilities. We are convinced that the

proposed process could be easily translated into machines. The catalytic approach is an additional advantage and is presented as a proof of concept, demonstrating that catalytic activities can be combined with separation and selectivity. A preliminary example of how it works is clearly depicted in Fig. 6h. Its purpose is to demonstrate that cages can offer the possibility of adding multifunctionality to the membranes from a chemistry and material design perspective. We are glad to see that Reviewer #2 is satisfied with our revision and responses [Comment 2].

[Comment 4] The authors claim in their response that they are highlighting the potential advantages of this membrane, but I don't see an annotated version of the manuscript or the changes described explicitly in the author response. That makes it hard to assess what changes were actually made.

Again in Reviewer 3, comment 1, the reviewer asks for context. The authors just say to refer to the new introduction but fail to explain what was changed.

Response: We are very sorry that we were not clear enough in the letter of "Response to the Editor and Reviewers". Regarding the Introduction section, the annotated revised part of the manuscript (introduction) is provided below. We are glad that also Reviewer 3 is now satisfied with the revision.

The annotated previous revision (relative to the first version) is below

~~“Highly permeable membranes with strict molecular/ion sieving are required for application in the Separation processes in the chemical and pharmaceutical industry to replace conventional separation processes such as evaporation, and distillation, and adsorption, which are energy intensive^{1, 2, 3}. Membrane technology does not necessarily involve thermal transitions and can be an effective alternative for molecular fractionation, and purification, particularly of systems sensitive to temperature. The selectivity of the membrane will dictate the number of steps required for an effective process. While commercial membranes have excellent performance in the separation of salts from water in large scale desalination plants, they lack in selectivity for molecules of similar size and charge. Multifunctional membranes are attractive for their additional capabilities such as photoresponsivity^{3, 4, 5}, pressure⁶, and thermal responsivity⁷. Besides high selectivity and permeance, and the perspective of multifunctional capacity, the potential of scaling up is important. A few methods have been established as competitive for the scaling up of~~

~~polymeric membranes.~~ Interfacial polymerization (IP) has been one of the most successful strategies for the fabrication of nanofiltration and reverse osmosis membranes^{4, 5, 6, 7, 8, 9}. By this method, an ultrathin selective layer is formed on a porous support, by reacting classical monomers such as m-phenylene diamine (MPD) dissolved in water in contact with solutions of acid chlorides (e. g. trimesoyl chloride, TMC) in a non-polar organic solvent, enabling the fabrication of large areas of membranes can be in this way fabricated with a relatively small quantity of monomers. Although densely crosslinked, there is random distribution of paths for transport with different sizes in the sub-nanometer range. This makes feasible and competitive the substitution of classical monomers such as m-phenylene diamine (MPD) by a variety of others with the perspective of providing a much better We believe that strict selectivity selectivity can only be achieved by using building blocks of preformed structure with precise free volume for selective permeation. With that in mind, our-

~~Our~~ group previously explored the fabrication of membranes by IP using macrocycles¹⁰ such as amine-functionalized cyclodextrin¹¹ and triethylamine¹² as monomeric units, recently further explored by Jiang et al.¹³ These highly crosslinked systems have much higher uniformity of “pores” for transport. In this work, we identify porous organic cages (POCs)^{14, 15, 16} synthesized by means of dynamic imine chemistry as a promising molecular platform for membrane design, with the perspective of providing even more selective and preformed tuned paths for permeation. The motivation is to profit from the highly defined 3D structures of the cages as building blocks, as potentially advantageous in comparison to the rather 2D units of macrocycles that we previously investigated.

Besides the tuned permeation paths, cages can offer the possibility of adding multifunctionality to the membranes. Multifunctional membranes could have photoresponsivity^{3, 17, 18}, pressure¹⁹, and thermal responsivity²⁰, and catalytic activity. We also reported smart covalent organic networks (CONs) with light switchable pores for molecular separation prepared by IP¹⁸. A key factor for the practical production of thin film composite (TFC) membranes is the reaction time. For production in continuous machines a fast reaction is essential. COF approaches are frequently based on slow dynamic Schiff base chemistry^{21, 22, 23, 24, 25, 26}; which might require more than 24 h to complete the film formation.

~~In the IP reaction conducted with a monomer like MPD in the aqueous phase and acryl chloride (e.g. trimesoyl chloride, TMC) in the organic phase, a ultrathin crosslinked polyamide membrane can be formed within 10 seconds on a porous polymer substrate. Other monomers are under investigation^{11, 12, 24, 25, 26}.~~ Multifunctional membranes are attractive for their additional capabilities

~~such as photoresponsivity^{3, 4, 5}, pressure⁶, and thermal responsivity⁷. Besides high selectivity and permeance, and the perspective of multifunctional capacity, the potential of scaling up is important. A few methods have been established as competitive for the scaling up of polymeric membranes. Other monomers are under investigation^{11, 12, 21, 22, 23}~~

~~We identify porous organic cages (POCs)^{27, 28, 29} synthesized by means of dynamic imine chemistry as a promising molecular platform for membrane design, preformed tuned paths for permeation can be created and their versatile structure could help to integrate additional functions. for multifunctionality purpose. We also reported smart covalent organic networks (CONs) with light-switchable pores for molecular separation prepared by IP¹⁸. A key factor for the practical production of thin film composite (TFC) membranes is the reaction time. For production in continuous machines a fast reaction is essential. COF approaches are frequently based on slow dynamic Schiff base chemistry^{24, 25, 26, 27, 28, 29}, which might require more than 24 h to complete the film formation.~~

~~Combining POC structures and interfacial polymerization fulfills the requirement of a fast reaction with intrinsic functionalities. POC properties have been fine-tuned for target-specific applications, such as bioimaging³⁰, rare gas recovery³¹, isotope hydrogen^{32, 33} and xylenes³⁴ and alkane/alkene separations³⁵ as adsorbents. Preliminary strategies...~~

Now relative to the previously revised version, the new changes in the introduction are annotated below in yellow.

Recently, a smart and responsive crystalline POC membrane was prepared by interfacial crystallization⁴⁰, showing a graded molecule sieving due to its switchable pore apertures. By a similar strategy, cages were formed and crystallized by counter-diffusion on porous anodic aluminum substrate to prepare membranes tested for ionic sieving with a superior selectivity for mono/divalent ions⁴¹. These results confirmed the potential of POCs for function-customization membranes. However, POC formation and crystallization require a long time. Furthermore, the scale-up of defect-free POCs membranes prepared by crystallization would be highly challenging. Interfacial polymerization has been recently performed with a mixture of cages and piperazine as a monomer for membrane fabrication tested for salts and dyes filtration⁴². However, as in the case of mixed matrix membranes approaches using mixtures of monomers dilute the advantages that POC would bring to a membrane.

[Comment 5] The authors seemed to miss the point of Reviewers 3 comment 4. This was meant to suggest that the paper belongs in a membrane journal, and not Nature Communications. I'm not convinced that the authors have made a compelling argument

that it belongs in this specific journal. It definitely should be published somewhere, though, but I continue to believe a membrane or separations journal is more appropriate.

Response: Based on our responses and revised manuscript, and considering that Reviewer 3 is satisfied, we believe this manuscript will be of general interest to the readership, with a focus on porous organic cages, engineered into scalable membranes for molecular separation. It is multidisciplinary, covering organic chemistry, organic cages, membrane science, materials science, and catalysis.